# INFSPLIGN: INFERENCE-TIME SPATIAL ALIGNMENT OF TEXT-TO-IMAGE DIFFUSION MODELS

## ABSTRACT

Text-to-image (T2I) diffusion models generate high-quality images but often fail to capture the spatial relations specified in text prompts. This limitation can be traced to two factors: lack of fine-grained spatial supervision in training data and inability of CLIP text embeddings to encode spatial semantics. We introduce InfSplign, a training-free inference-time method that improves spatial alignment by adjusting the noise through a compound loss in every denoising step. Our proposed loss leverages different levels of cross-attention maps extracted from the U-Net decoder to enforce accurate object placement and a balanced object presence during sampling. Our method is lightweight, plug-and-play, and compatible with any diffusion backbone. Our comprehensive quantitative and qualitative evaluations demonstrate that, on widely adopted spatial benchmarks (VISOR and T2I-CompBench), our approach establishes a new state-of-the-art (to the best of our knowledge), delivering substantial performance gains and even surpassing fine-tuning-based baselines.

## 1 INTRODUCTION

Diffusion-based text-to-image (T2I) generative models have rapidly advanced, enabling the synthesis of high-quality, detailed images from arbitrary textual descriptions (Rombach et al., 2022; Saharia et al., 2022; Ho et al., 2022; Dhariwal & Nichol, 2021; Nichol & Dhariwal, 2021; Chang et al., 2023; Podell et al., 2024; Balaji et al., 2022). Despite these developments, precise control over spatial relationships described in text prompts remains challenging, manifesting as misplacement or unintended merging of objects in generated images or even completely failing to depict all specified objects or attributes (Gokhale et al., 2022). Diffusion models frequently fail to distinguish between prompts such as *"object A to the left of object B"* and *"object A to the right of object B"*, often producing nearly identical outputs irrespective of spatial cues as shown in Fig. 1. This misalignment substantially reduces reliability, hindering applications that demand accurate spatial reasoning, such as generating scene layouts for robotic manipulation and visual grounding in augmented reality systems (Chen et al., 2024a). Beyond misplacement, diffusion models frequently drop objects altogether, or allow one object to dominate, erasing the other. This undermines reliability in compositional generation, where preserving all entities is as important as placing them correctly. The deficiency in spatial understanding is quantitatively evident; for instance, on the T2I-CompBench (Huang et al., 2023) compositional reasoning benchmark, state-of-the-art performances on spatial understanding are around 20%, significantly lagging behind performance on other aspects such as attribute binding (around 50%). This performance gap underscores a critical area within T2I research, emphasizing the importance of developing solutions that effectively address spatial misalignment.

Approaches tackling spatial accuracy or object preservation broadly fall into two categories: *fine-tuning-based* and *inference-time* methods. Fine-tuning-based methods typically employ spatially-aware datasets, auxiliary reward models, or explicit training mechanisms to enforce spatial accuracy (Feng et al., 2023; Zhang et al., 2024a), achieving relatively high spatial accuracy at the cost of considerable computational overhead and the risk of negatively impacting the carefully optimized diffusion backbone and its generalizability. In contrast, inference-time methods (Voynov et al., 2023; Epstein et al., 2023) avoid expensive re-training, providing computationally efficient alternatives capable of flexible spatial adjustments during sampling. However, current inference-time methods remain overly complex, relying on auxiliary inputs like layout maps (Sun et al., 2024), scene graphs (Farshad et al., 2023), or external guidance from large language models (LLMs) (Phung et al., 2024; Lian et al., 2024), thereby limiting ease of deployment and interoperability.

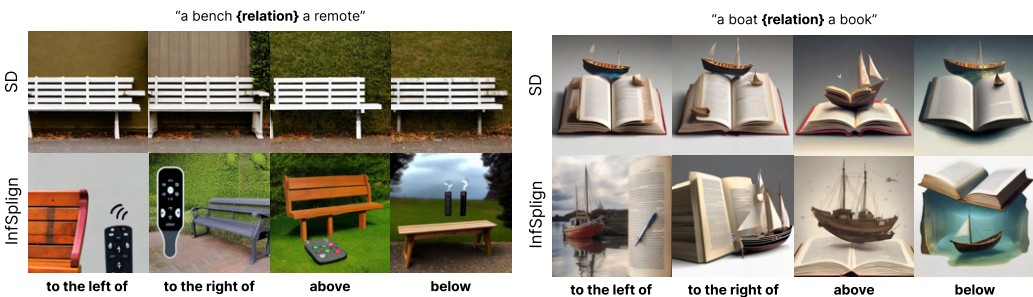

Figure 1: `InfSplign` is a training-free inference-time light-weight method that improves spatial understanding of text-to-image (T2I) Stable Diffusion (SD) models.

A hypothesis is that the spatial limitations arise from the CLIP text encoder (Radford et al., 2021), commonly used in pretrained diffusion models, that fails to adequately encode spatial semantics (Gokhale et al., 2022). Other studies have hypothesized that classifier-free guidance (CFG) might be entangling multiple semantic factors in the text prompt (Wu & la Torre, 2024), and that "positive" prompt might be too weak to enforce spatial alignment (Chang et al., 2024), proposing a contrastive setting with both positive and negative prompts. Another line of research investigates the impact of the quantity of training data exhibiting spatial relationships on model performance; for *e.g.*, SPRIGHT (Chatterjee et al., 2024b) generates new spatially-focused captions for four widely used datasets and finetunes the model on them. However, fine-tuning introduces significant challenges, particularly regarding computational efficiency, scalability to large models, and the risk of degrading or interfering with the model's original pre-trained capabilities (*e.g.*, through catastrophic forgetting or reduced generalization).

Instead of introducing additional training complexity or external data inputs, we explore a computationally efficient and scalable approach leveraging information already present within the diffusion process. We focus on directly extracting spatial information from attention maps during the early stages of reverse diffusion (generation) to guide the sampling process, and demonstrate that attention maps can serve as proxies for spatial information. Building on this insight, we introduce `InfSplign`, an inference-time method that partitions the U-Net decoder cross-attention maps into three hierarchical levels: coarse, mid-level, and fine-grained. From the coarse and mid-level attentions, we extract object centroids and variances, which are then used to define three complementary loss terms. Together, these losses enforce spatial alignment and promote balanced object representations in the final denoised outputs. At each denoising step, the compound loss is applied to refine the predicted noise for the subsequent timestep, thereby guiding the sampling trajectory toward spatially coherent images without altering the model parameters. Through extensive ablations and benchmark evaluations, we demonstrate that this minimal but targeted intervention results in improving spatial alignment of T2I models. Our core contributions can be summarized as follows:

- *Spatial Alignment and Object Preservation.* We introduce `InfSplign`, a training-free inference-time approach that leverages U-Net attention maps to enforce object spatial alignment and preservation. Our compound loss is comprised of three components. In addition to an object location loss that enforces accurate spatial grounding, we incorporate an object presence loss, which increases the certainty of object representation, and an object balance loss, which mitigates cross-object interference in the finer layers of the U-Net decoder.
- *Experimental Results.* Extensive evaluations on spatial benchmarks, including VISOR and T2I-CompBench, demonstrate that `InfSplign` improves spatial alignment by up to 24.81% and 21.91% over state-of-the-art inference-time methods, and even surpasses fine-tuning approaches by 14.33% and 9.72%. Extensive ablation studies and qualitative results further corroborate the effectiveness of our proposed approach.

## 2  RELATED WORK

**Text-to-Image (T2I) Generation** aims to produce visually realistic images that align with natural language prompts. While earlier work focused on GANs (Isola et al., 2017; Li et al., 2022; Park et al., 2019) and autoregressive models (Ramesh et al., 2021; Yu et al., 2022), diffusion models (Ho et al., 2020; Rombach et al., 2022; Saharia et al., 2022) have become the dominant approach due to their superior image fidelity, diversity, and stability (Liu et al., 2022). The integration of vision–language

pretraining strategies such as CLIP (Radford et al., 2021) helped further enhance semantic alignment (Ramesh et al., 2022), yet recent studies show that even state-of-the-art models struggle with accurately capturing fine-grained textual details, particularly spatial relationships (Huang et al., 2023; Wang et al., 2025). To address this, expansion of network architectures or new training objectives have been explored (Feng et al., 2023; Li et al., 2024), but these require costly retraining.

**Spatial Understanding and Object Preservation in T2I Models.** Efforts aimed at improving spatial understanding primarily fall into fine-tuning-based or layout-conditioned methods. Fine-tuning approaches enhance spatial reasoning by training models on spatially-aware datasets or using auxiliary objectives, such as reward-based optimization (Zhang et al., 2024c; Chen et al., 2023; Zhang et al., 2024b). SPRIGHT (Chatterjee et al., 2024b) presents a large scale vision-language dataset for fine-tuning diffusion model for spatial data. CoMPaSS (Zhang et al., 2024a) advances state-of-the-art spatial accuracy on common benchmarks by explicitly incorporating spatially labeled data during training. However, these methods involve expensive retraining processes and risk destabilizing the pretrained diffusion backbone. Another category explicitly injects spatial layout information (sometimes through LLMs), such as bounding boxes, depth maps, or segmentation masks, to guide generation (Sun et al., 2024; Chen et al., 2024b; Gong et al., 2024; Phung et al., 2024; Li et al., 2023a; Lee & Sung, 2024; Nie et al., 2024; Lian et al., 2024; Derakhshani et al., 2023). However, these methods depend on external layout inputs which may not always be available, thus limiting usability, and require additional pre-processing and computational overhead.

**Inference-Time Guidance for Diffusion Models** circumvent costly retraining by directly manipulating diffusion processes during sampling. Methods like Attend & Excite (Chefer et al., 2023) address the issue of missing objects by optimizing attention maps at inference time, but do not explicitly enforce spatial accuracy. Structured Diffusion Guidance (Feng et al., 2023) manipulates attention maps for improved layout control, yet lacks explicit modeling of spatial relationships described by textual prompts. Composable Diffusion (Liu et al., 2022) interprets diffusion models as energy-based compositions of individual concepts, improving object presence but providing minimal spatial control. More targeted spatial inference-time methods, such as Prompt-to-Prompt (Hertz et al., 2023) and DIVIDE&BIND (Li et al., 2023b), demonstrate the potential of directly modifying cross-attention maps. Recent information-theoretic insights further motivate inference-time interventions: analysis of mutual information between text prompts and images (Wang et al., 2025; Kong et al., 2024), initial diffusion noise predetermines object layout generation (Ban et al., 2024), and sometimes needs to be guided to produce a valid sample (Guo et al., 2024) according to the prompt. Diffusion Self-Guidance (Epstein et al., 2023) develops a framework for image editing by controlling the appearance, shape, size and location of objects but limits sample diversity. REVISION (Chatterjee et al., 2024a) generates spatially accurate synthetic images as conditional input, reducing the task to an image-to-image (I2I) pipeline. STORM (Han et al., 2025) introduces a distribution-based loss using Optimal Transport (OT) (Villani et al., 2008) to adjust attention maps toward target distributions fixed at specified spatial relationships. Unlike prior approaches which rely on external targets or synthetic images, `InfSplign` directly regulates object preservation during sampling, ensuring both alignment and completeness. To our knowledge, no prior inference-time method uses attention variance as a principled measure of object certainty.

## 3 INFSPLIGN: INFERENCE-TIME SPATIAL ALIGNMENT

### 3.1 PRELIMINARIES

**Diffusion Models** provide an effective framework for sampling from complex data distributions $q(x)$ by learning to invert a forward diffusion process. The forward process is a Markov chain that iteratively adds Gaussian noise to a clean data point $x_0 \in \mathcal{X}$ over $T$ steps: $x_t = \sqrt{\bar{\alpha}_t}\, x_0 + \sqrt{1 - \bar{\alpha}_t}\, \epsilon_t$, where $\epsilon_t \sim \mathcal{N}(0, I)$, and $\{\beta_t\}_{t=1}^T$ is a variance schedule with $\alpha_t = 1 - \beta_t$ and $\bar{\alpha}_t = \prod_{i=1}^t \alpha_i$ (Ho et al., 2020; Nichol & Dhariwal, 2021). The reverse process defines a generative model $p_\theta(x_{t-1}|x_t)$ that approximates the true posterior $q(x_{t-1}|x_t, x_0)$. A neural network $\epsilon_\theta(x_t, t)$, typically a U-Net (Ronneberger et al., 2015), is trained to predict the noise added during the forward process. At inference time, a simplified update rule can be written as $x_{t-1} \approx x_t - s_t\, \epsilon_\theta(x_t, t)$, where $s_t$ is a step-size factor depending on the variance schedule. A conditioning variable $y$ (in our case, text prompts) can be incorporated, resulting in conditional predictions $\epsilon_\theta(x_t, t, y)$ (Zhang et al., 2023a; Mo et al., 2023). Our work is built upon a text-conditioned latent diffusion model, Stable Diffusion (Rombach et al., 2022), which operates in latent space $z_t$ produced by a pretrained autoencoder.

**Inference-Time Guidance of Diffusion Models.** Diffusion models can be adapted to a wide range of downstream tasks at inference time (without retraining or fine-tuning) through input conditioning (*e.g.* text prompts) and external reward modes (*e.g.* CLIP-based scores), to influence the denoising strategy to better align with desired outcomes. Classifier guidance (Dhariwal & Nichol, 2021) steers generation using gradients from a pretrained image classifier, whereas classifier-free guidance (Ho et al., 2021) eliminates the need for an external classifier by training the model to denoise both with and without conditioning, and then interpolating between the two at inference time. Diffusion models can also be interpreted through their score-based formulation, where the model estimates the gradient of the log probability density, $\nabla_{z_t} \log p(z_t, t)$. This gives us an intuition about the direction to move in to increase the log likelihood of our data sample based on some conditional information. While the denoising formulation, $\epsilon_\theta(z_t, t)$, gives us a prediction of the noise that was added by the forward diffusion model at each timestep, the inference-time classifier-free guidance (CFG) approach (Ho et al., 2021) is commonly adopted to guide a conditional reverse diffusion process toward a desired conditioning signal (a text prompt $y$ in our T2I setting). The score-based formulation of CFG is comprised of two conditional and unconditional terms:

$$\nabla_{z_t} \log p(z_t|y, t) \approx \nabla_{z_t} \log p(z_t, t) + \gamma(\nabla_{z_t} \log p(z_t|y, t) - \nabla_{x_t} \log p(z_t, t)), \tag{1}$$

where $\gamma$ is the guidance strength. The equivalent noise prediction form at step $t$ can be given by:

$$\epsilon_t = \epsilon_\theta(z_t; t) + \gamma(\epsilon_\theta(z_t; t, y) - \epsilon_\theta(z_t; t)). \tag{2}$$

## 3.2 A Deeper Dive into InfSplign

Our work is focused on inference-time guidance, and serves as a lightweight, plug-and-play enhancement to diffusion models. To tackle with data and caption limitations, we introduce a guidance signal/loss to quantify the misalignment between the generated latent attention maps and the spatial cues in the prompt, as well as to ensure a balanced representation of all objects throughout the reverse diffusion process. The idea is to *actively nudge* the generation process towards generating more spatially-cognizant images. Fig. 2 gives a high-level overview of the mechanics of InfSplign. The input to the system is a user-specified text prompt, *e.g. "a potted plant to the right of a clock"*, and the noisy latent embedding $z_t$ produced at timestep $t$ of the reverse diffusion process. We represent the prompt as a structured triplet - $\langle A, R, B \rangle$, where $A$ and $B$ are the object tokens and $R \in \mathcal{R}$, where $\mathcal{R}$ is the set of spatial relationships, e.g. $A=$"potted plant", $B=$"clock" and $R=$"to the right of".

To guide the denoising process, we extract the cross-attention maps corresponding to the object tokens $A$ and $B$ at timestep $t$. We divide the attention maps into three abstraction levels: (i) **coarse attention:** where presence of the objects is shaped; (ii) **mid-level attention:** at which we observed objects might dominate each other in

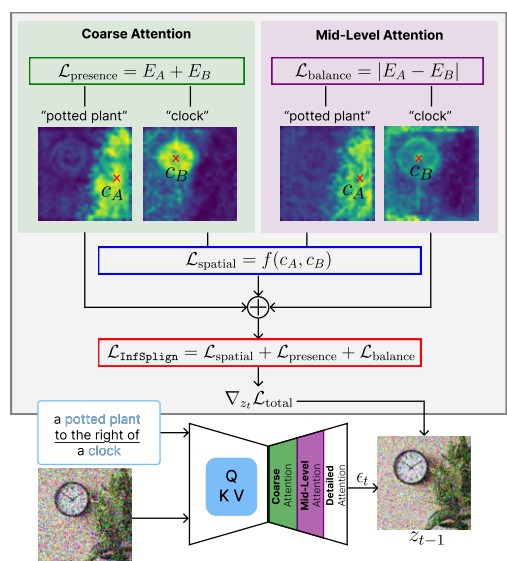

Figure 2: Overview of our approach: At each timestep, InfSplign extracts object-token attention maps from coarse and mid-level attention layers, computes centroids $c_1$ and $c_2$, and applies $\mathcal{L}_{\text{spatial}}$ on their relative positions. Coarse attention yields $\mathcal{L}_{\text{presence}}$ to preserve all objects and mid-level attention provides $\mathcal{L}_{\text{balance}}$ to prevent object dominance. The gradient of $\mathcal{L}_{\text{InfSplign}}$ adjusts the noise before the next denoising step.

magnitude as such impacting their representation in the final outcome; (iii) **fine-grained attention:** where nuances and high-resolution subtleties of the final image appear. We focus on the first two levels (coarse and mid), where spatial cognizance of the model is formed. As we elaborate later on, our loss is comprised of three terms: **spatial alignment loss** ($\mathcal{L}_{\text{spatial}}$), **object presence loss** ($\mathcal{L}_{\text{presence}}$), **representation balance loss** ($\mathcal{L}_{\text{balance}}$) constituting our proposed total loss $\mathcal{L}_{\text{InfSplign}}$. Both coarse and mid-level attention maps are used to estimate spatial alignment loss. We utilize the coarse-level attention layers for the presence loss to minimize the variance of each object's attention map, thus

ensuring that all objects will be maintained through the reverse diffusion steps. We use the mid-level attention layers in our representation balance loss to avoid any of the objects over-shadowing the other one from the magnitude perspective. The gradient of $\mathcal{L}_{\texttt{InfSplign}}$ with respect to (w.r.t.) the latent $z_t$ provides a guidance signal that modifies the predicted noise such that the latent is shifted in a direction leading to a more spatially aligned image:

$$\epsilon_t \leftarrow \epsilon_\theta(z_t; t) + \gamma(\epsilon_\theta(z_t; t, y) - \epsilon_\theta(z_t; t)) + \eta \nabla_{z_t} \mathcal{L}_{\texttt{InfSplign}} = \epsilon_t + \nabla_{z_t} \mathcal{L}_{\texttt{InfSplign}}, \quad (3)$$

where $\eta$ is a weight parameter acting similar to guidance strength ($\gamma$), balancing the magnitude of the last two terms. This updated noise prediction is used to compute the denoised latent $z_{t-1}$ guiding the generation towards spatially cognizant images. Thus, the final update direction combines both semantic (via CFG) and overall spatial alignment (imposed via $\mathcal{L}_{\texttt{InfSplign}}$). This procedure is applied iteratively over all reverse diffusion timesteps.

**From Attention to Centroids and Variances.** It is demonstrated in (Hertz et al., 2023) that cross-attention layers encode rich information about the spatial arrangement of objects in generated images. Building on this insight, we extract attention maps from coarse- and mid-level decoder layers of the U-Net, which most reliably capture object structure and spatial location. We estimate the position of each object by computing the centroid of its attention distribution. The centroid $c_A$ of token $A$ is computed as a weighted average over the spatial coordinates of the latent $z_t$, using the attention layer $l$ weights of token $A$ (denoted by $\mathcal{A}_t^{(l)}$) as coefficients and normalized by the total attention mass:

$$c_A^{(l)} = (x_A, y_A) = \left( \frac{\sum_{h,w} \mathcal{A}_t^{(l)}[h, w] \cdot x_w}{\sum_{h,w} \mathcal{A}_t^{(l)}[h, w]}, \quad \frac{\sum_{h,w} \mathcal{A}_t^{(l)}[h, w] \cdot y_h}{\sum_{h,w} \mathcal{A}_t^{(l)}[h, w]} \right), \quad (4)$$

where $h \in [H]$ and $w \in [W]$ scan over the height and width of the corresponding attention map of size $H \times W$, where the origin, $(1, 1)$, is assumed to be on the top-left of the tensor. Note that since the centroids are derived from U-Net cross-attention maps, which themselves depend on the latent representation $z_t$, the resulting loss function in Eq. 8 remains differentiable w.r.t. $z_t$. Additionally, we incorporate variance as a measure of uncertainty in the attention maps. Low variance indicates high attention values close to the centroid and low attention away from it, thus encouraging distinct object representations, while high variance suggests weak attention, risking object omission. So, we compute the variance $\sigma_A^{2(l)}$ of the attention layer $l$ distribution of token $A$ by weighting the squared distance of each pixel from the centroid $c_A$ (in Eq. 4) by its attention value, normalized by the total attention mass:

$$\sigma_A^{2(l)} = \left( \frac{\sum_{h,w} \mathcal{A}_t^{(l)}[h, w] \cdot \|(x_w, y_h) - c_A^{(l)}\|^2}{\sum_{h,w} \mathcal{A}_t^{(l)}[h, w]} \right). \quad (5)$$

**Spatial Alignment Loss.** The spatial relationship $R$ between two objects $A$ and $B$ is expressed as a difference between their centroids, denoted as $\Delta$, and is computed along the appropriate axis as:

$$\Delta = \begin{cases} x_B - x_A \text{ for "left"}, & x_A - x_B \text{ for "right"}, \\ y_B - y_A \text{ for "above"}, & y_A - y_B \text{ for "below"}, \quad \|x_A - x_B\| \text{ for "near"}. \end{cases} \quad (6)$$

This captures the directional alignment between the two objects and signals adherence to the specified spatial relation. Building on this, we define the spatial loss function as:

$$\mathcal{L}_{\text{spatial}} = f_{\text{spatial}}(\alpha(m - \Delta)), \quad (7)$$

where $f_{\text{spatial}}$ can be any function from the $\texttt{ReLU}$ family of losses, $m$ is a distance margin indicating the acceptable minimum distance between the objects' centroids, and $\alpha$ is a scaling factor controlling the steepness of the loss. Notably, $\alpha$ sharpens the slope around the decision boundary, determining how strictly the model is penalized. $\mathcal{L}_{\text{spatial}}$ results in a small penalty if the objects are placed correctly w.r.t. the target spatial relation, *e.g.* if $\mathcal{S} = $ *"to the right of"*, $x_A - x_B > m$, $\mathcal{L}_{\text{spatial}} \to 0$. The loss is high when objects violate the spatial relation, or are too close to each other, or $\Delta < m$.

**Object Presence Loss.** A prerequisite of spatial alignment is to ensure both objects remain visible in the final denoised image. We approach this problem from a variance-based perspective and interpret the attention weights as an *energy distribution* over the spatial location. As illustrated in Fig. 3, attention energy for a token gradually concentrates into a single region during the diffusion process. This indicates that uncertainty in token localization decreases over time. We hypothesize that insufficient certainty in an object's attention distribution leads to its omission or weak representation in the final image. To address this, we introduce an object presence loss, $\mathcal{L}_{\text{presence}} = \sigma_A^{2(l)} + \sigma_B^{2(l)}$, that minimizes the variance of each object's attention map in layer $l$.

**Representation Balance Loss.** Another common failure case is the omission of an object due to a weaker representation in the attention map, e.g. $\sigma_A \gg \sigma_B$ causes object $A$ to dominate object $B$. At coarse-level the attention layers encode global information, so, this imbalance is less problematic. However, the mid-level attention layers might suppress the weaker object. To mitigate this, we enforce that both objects maintain comparable levels of uncertainty. As shown in Fig. 3, these coarse attention maps often capture clusters of

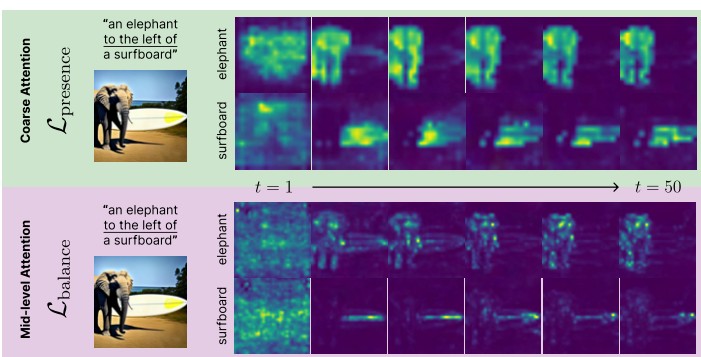

Figure 3: Attention energy across U-Net decoder cross-attention layers. Coarse layers encode global layout, so we keep attention concentrated. Mid-level layers capture details so we balance object energies to prevent overshadowing.

fine-grained details. In such layers, simply minimizing variance would undesirably restrict the model's ability to explore object details. Instead, we ensure parity of uncertainty across objects by introducing an object balance loss, $\mathcal{L}_{\text{balance}} = |\sigma_A^{2(l)} - \sigma_B^{2(l)}|$, which encourages both objects to have a similar degree of dispersion in their attention maps in layer $l$. This prevents one object from overshadowing the other, ensuring a balanced representation.

We arrive at our proposed spatial alignment loss (coined as $\mathcal{L}_{\texttt{InfSplign}}$) by jointly enforcing correct placement, object preservation and a balanced representation of objects, with hyperparameters $\lambda_{\text{s}}$, $\lambda_{\text{p}}$, and $\lambda_{\text{b}}$ controlling the relative strength of each component:

$$\mathcal{L}_{\texttt{InfSplign}} = \lambda_{\text{s}}\mathcal{L}_{\text{spatial}} + \lambda_{\text{p}}\mathcal{L}_{\text{presence}} + \lambda_{\text{b}}\mathcal{L}_{\text{balance}}. \qquad (8)$$

Unlike Attend-and-Excite (Chefer et al., 2023), which implicitly enforces object presence by maximizing attention energy (often at the expense of other tokens) or STORM (Han et al., 2025), which fixes the object locations (unnecessary overparameterization) to achieve adherence with the spatial cues in the text prompt, our proposed preserves generation diversity by penalizing misplaced or omitted objects through gradients computed w.r.t the latent $z_t$ during sampling. The pseudocode for a single reverse diffsion step under InfSplign is in Alg. 1.

---

**Algorithm 1:** Denoising with `InfSplign`

**Require:** $\mathcal{P} = \langle A, R, B \rangle, z_t, \texttt{SD}(.), \eta$
1  $\mathcal{A}_t, \mathcal{B}_t, \epsilon_t \leftarrow \texttt{SD}(z_t, \mathcal{P})$
2  $c_A, c_B \leftarrow \texttt{Centroid}(\mathcal{A}_t, \mathcal{B}_t)$ using Eq. 4
3  $\sigma_A^2, \sigma_B^2 \leftarrow \texttt{Variance}(\mathcal{A}_t, \mathcal{B}_t, c_A, c_B)$ using Eq. 5
4  $\Delta \leftarrow \texttt{Difference}(c_A, c_B, R)$ using Eq. 6
5  $\mathcal{L}_{\text{spatial}} \leftarrow f_{\text{spatial}}(\Delta, \alpha, m)$ using Eq. 7
6  $\mathcal{L}_{\text{presence}} \leftarrow \sigma_A^2 + \sigma_B^2, \mathcal{L}_{\text{balance}} \leftarrow |\sigma_A^2 - \sigma_B^2|$
7  $\mathcal{L}_{\texttt{InfSplign}} = \lambda_{\text{s}}\mathcal{L}_{\text{spatial}} + \lambda_{\text{p}}\mathcal{L}_{\text{presence}} + \lambda_{\text{b}}\mathcal{L}_{\text{balance}}$
8  $\epsilon_t \leftarrow \epsilon_t + \eta \cdot \nabla_{z_t}\mathcal{L}_{\texttt{InfSplign}}$ using Eq. 3
9  $z_{t-1} \leftarrow z_t - s_t \cdot \epsilon_t$
**Return:** $z_{t-1}$

---

## 4 EXPERIMENTS

We present a comprehensive evaluation showing the efficacy of our approach against state-of-the-art baselines. Next, we conduct detailed ablation studies to assess the impact of the key (hyper)parameters. Finally, we showcase qualitative results highlighting the superior performance of `InfSplign`.

**Implementation Details.** We apply `InfSplign` on top of SD v1.4 and v2.1 models (Rombach et al., 2022) for 50 inference steps, which is standard protocol adopted by prior work (Han et al., 2025; Chatterjee et al., 2024a). The hyperparameters $\alpha$ and margin are set to $\alpha = 1.5$, $m = 0.25$ (SD v1.4), $m = 0.5$ (SD v2.1) respectively, and $f_{\text{spatial}}(.)$ to GeLU (Hendrycks & Gimpel, 2016), CFG guidance scale $\gamma = 7.5$, and the guidance weight for our loss is set to $\eta = 1000$ to balance magnitudes. For *coarse attention* and *mid-level attention*, we use the cross-attention layers $1 - 3$ of the first and second blocks of the U-Net decoder, respectively. The $\mathcal{L}_{\text{presence}}$ loss is computed from the attention maps of the first block, while $\mathcal{L}_{\text{balance}}$ is derived from those of the second block. Hyperparameters are selected via grid search (see Appendix B), yielding $\lambda_{\text{s}}$=0.5, $\lambda_{\text{p}}$=1, $\lambda_{\text{b}}$=0.5 for SD v1.4 and $\lambda_{\text{s}}$=0.5, $\lambda_{\text{p}}$=1, $\lambda_{\text{b}}$=1.0 for SD v2.1. We use the same random seeds for each benchmark as defined in the original papers.

Table 1: **Performance comparison between different models on VISOR (%) and Object Accuracy (OA) (%) metrics**, based on Stable Diffusion 1.4, and 2.1. ♣ reported from (Han et al., 2025)

| Model | Venue | Fine-tuning | Extra Input | OA (%) | VISOR (%) | | | | | |
|---|---|---|---|---|---|---|---|---|---|---|
| | | | | | uncond | cond | 1 | 2 | 3 | 4 |
| **Stable Diffusion 1.4** | | | | | | | | | | |
| SD 1.4 (Rombach et al., 2022) | | ✗ | ✗ | 29.86 | 18.81 | 62.98 | 46.60 | 20.11 | 6.89 | 1.63 |
| SD 1.4 + CDM (Liu et al., 2022) | ECCV22 | ✓ | ✗ | 23.27 | 14.99 | 64.41 | 39.44 | 14.56 | 4.84 | 1.12 |
| GLIDE (Nichol et al., 2022) | ICML22 | ✓ | ✗ | 3.36 | 1.98 | 59.06 | 6.72 | 1.02 | 0.17 | 0.03 |
| GLIDE + CDM (Liu et al., 2022) | ECCV22 | ✓ | ✗ | 10.17 | 6.43 | 63.21 | 20.07 | 4.69 | 0.83 | 0.11 |
| Control-GPT (Zhang et al., 2023b) | arXiv23 | ✓ | ✓ | 48.33 | 44.17 | 65.97 | 69.80 | 51.20 | 35.67 | 20.48 |
| CoMPaSS (Zhang et al., 2024a) | ICCV25 | ✓ | ✗ | 65.56 | 57.41 | 87.58 | 83.23 | 67.53 | 49.99 | 28.91 |
| Structure Diffusion (Feng et al., 2023) | ICLR23 | ✗ | ✗ | 28.65 | 17.87 | 62.36 | 44.70 | 18.73 | 6.57 | 1.46 |
| Attend-and-Excite (Chefer et al., 2023) | SIGGRAPH23 | ✗ | ✗ | 42.07 | 25.75 | 61.21 | 49.29 | 19.33 | 4.56 | 0.08 |
| Divide-and-Bind♣ (Li et al., 2023b) | BMVC24 | ✗ | ✗ | 46.03 | 31.62 | 68.70 | 64.72 | 37.82 | 18.64 | 5.30 |
| INITNO♣ (Guo et al., 2024) | CVPR24 | ✗ | ✗ | 60.40 | 35.18 | 58.24 | 71.20 | 42.71 | 20.09 | 6.72 |
| Layout Guidance (Chen et al., 2024b) | WACV24 | ✗ | ✓ | 40.01 | 38.80 | 95.95 | - | - | - | - |
| CONFORM♣ (Meral et al., 2024) | CVPR24 | ✗ | ✗ | 60.73 | 38.48 | 62.33 | 73.01 | 45.82 | 25.57 | 9.52 |
| REVISION (Chatterjee et al., 2024a) | ECCV24 | ✗ | ✓ | 53.96 | 52.71 | 97.69 | 77.79 | 61.02 | 44.90 | 27.15 |
| STORM (Han et al., 2025) | CVPR25 | ✗ | ✗ | 61.01 | 57.58 | 94.39 | 85.93 | 69.71 | 49.01 | 25.70 |
| **InfSplign (Ours)** | | ✗ | ✗ | **67.36** | **66.54** | **98.79** | **90.48** | **77.79** | **61.37** | **36.54** |
| Improvement (Inference Time) | | | | +6.35 | +8.96 | +1.10 | +4.55 | +8.08 | +12.36 | +9.39 |
| Improvement (All) | | | | +1.80 | +8.96 | +1.10 | +4.55 | +8.08 | +11.38 | +7.63 |
| **Stable Diffusion 2.1** | | | | | | | | | | |
| SD 2.1 (Rombach et al., 2022) | | - | ✗ | 47.83 | 30.25 | 63.24 | 64.42 | 35.74 | 16.13 | 4.70 |
| SPRIGHT (Chatterjee et al., 2024b) | ECCV24 | ✓ | ✗ | 60.68 | 42.23 | 71.24 | 71.78 | 51.88 | 33.09 | 16.15 |
| REVISION (Chatterjee et al., 2024a) | ECCV24 | ✗ | ✓ | 48.26 | 47.11 | 97.61 | 76.07 | 55.75 | 37.10 | 19.53 |
| STORM (Han et al., 2025) | CVPR25 | ✗ | ✗ | 62.55 | 59.35 | 94.88 | 88.34 | 71.75 | 52.03 | 25.42 |
| CoMPaSS (Zhang et al., 2024a) | ICCV25 | ✓ | ✗ | 68.22 | 62.06 | 90.96 | 85.02 | 71.29 | 56.03 | 35.90 |
| **InfSplign (Ours)** | | ✗ | ✗ | **77.28** | **76.26** | **98.68** | **94.65** | **86.66** | **73.48** | **50.23** |
| Improvement (Inference Time) | | | | +14.73 | +16.91 | +1.07 | +6.31 | +14.91 | +21.45 | +24.81 |
| Improvement (All) | | | | +9.06 | +14.20 | +1.07 | +6.31 | +14.91 | +17.45 | +14.33 |

**Baselines and Benchmarks.** We compare our approach with the most relevant and recent baselines, both *inference-time* (Han et al., 2025; Chatterjee et al., 2024a; Chefer et al., 2023; Feng et al., 2023; Li et al., 2023b; Guo et al., 2024; Meral et al., 2024) and *fine-tuning* (Chatterjee et al., 2024b; Zhang et al., 2024a; Liu et al., 2022). As InfSplign is an inference-time method, a fair comparison would be against inference-time approaches, yet InfSplign even outperforms fine-tuning approaches. We evaluate our performance on the two most commonly adopted benchmarks assessing spatial understanding of diffusion models, namely VISOR (Gokhale et al., 2022) and T2I-CompBench (Huang et al., 2023). Notably, VISOR is focused on evaluating how well T2I models generate correct spatial relationships described in the text prompt, whereas T2I-CompBench assesses T2I models on broader compositional metrics such as attribute binding, numeracy, and complex compositions, with a module dedicated to object relations pertinent to our evaluation angle.

**Quantitative Evaluation: VISOR.** Tab. 1 summarizes the VISOR scores of our method compared to relevant approaches in T2I space. As can be seen, InfSplign consistently outperforms both fine-tuning and inference-time methods across all baselines by a significant margin. On SD v1.4, InfSplign surpasses the best of inference-time methods (such as CONFORM, INITNO, and STORM), as well as fine-tuning-based methods (e.g., SPRIGHT and CoMPaSS). We achieve the highest score across all metrics. Most notably on object accuracy (OA), we outperform the best of inference-time and fine-tuning based competitors by $6.35\%$ and $1.8\%$, respectively. The same holds for the most challenging task of the benchmark, i.e. "unconditional" score, with $8.96\%$ and $9.13\%$, and VISOR-4, with $9.39\%$ and $7.63\%$, respectively. This demonstrates that InfSplign can surpass the performance of models that require additional input and/or retraining. These results indicate that when both objects are successfully generated, our method ensures their spatial relationship is correct. This trend continues to further improve with the stronger backbones SD v2.1 consistently. The margin is significantly larger, where we report up to $24.81\%$ and $14.33\%$ score improvement on the challenging VISOR-4 task compared to the state-of-the-art inference-time and fine-tuning baselines, STORM and ComPaSS, respectively. This highlights the consistency of our method in generating 4 out of 4 spatially correct images across all test prompts. One key factor behind these substantial improvements is the higher quality of attention maps with stronger backbones, as our method relies on attention maps to guide the denoising process.

**Quantitative Evaluation: T2I-CompBench.** Tab. 2 summarizes our performance compared to the state-of-the-art baselines both inference-time and fine-tuning on SD v1.4 and v2.1. Here again, `InfSplign` consistently outperforms competitors by a margin. On SD v1.4, it improves over the leading inference-time method REVISION by $+4.31\%$, while REVISION requires *additional inputs*. Therefore, a more direct comparison is with STORM, in which `InfSplign` offers a significant $+21.58\%$ margin. Furthermore, our method even surpasses the strongest fine-tuning baseline in this setting, CoMPaSS, by $+3.71\%$. Following the same trend as in Tab. 1, with the stronger backbone SD v2.1 our improvement margin increases significantly to $+21.91\%$ and $+9.72\%$ respectively.

**Ablation Studies.** To better investigate the impact of the main (hyper)parameters and analyze the behavior of our loss functions, we conduct extensive studies. For this purpose, we use a subset of VISOR, which we refer to as $VISOR_{3160}$, in which instead of all 4 spatial relations $R$ between two objects $A$ and $B$, we randomly sample a single relation. This results in 3160 prompts that still cover all objects while keeping the dataset size manageable for ablations. Firstly, we perform a grid search over the key hyperparameters $\alpha$, $m$, $\lambda_s$, $\lambda_p$, and $\lambda_b$ for SD v1.4 and v2.1. The results of this extensive study are summarized in Tab. 4 and Tab. 5, in Appendix B. The optimal values obtained from this study are subsequently used for the experiments on the complete VISOR benchmark reported earlier in Tab. 1 and Tab. 3, and the T2I-CompBench results in Tab. 2.

Table 2: Performance summary on T2I-CompBench on SD v1.4 and SD v2.1 backbone. **FT**: fine-tuning, **EI**: extra inputs (EI). Best results are in **bold**.

| Method | Venue | FT | EI | **SD1.4** | **SD2.1** |
|---|---|---|---|---|---|
| Baseline (2022) | | - | - | 0.1246 | 0.1342 |
| Comp. Diff. (2022) | ECCV22 | ✓ | ✗ | - | 0.0800 |
| Struc. Diff. (2023) | ICLR23 | ✗ | ✗ | - | 0.1386 |
| Att.&Ex. (2023) | SGRAPH23 | ✗ | ✗ | - | 0.1455 |
| SPRIGHT (2024b) | ECCV24 | ✓ | ✗ | - | 0.2133 |
| Revision (2024a) | ECCV24 | ✗ | ✓ | 0.3340 | - |
| CoMPaSS (2024a) | ICCV25 | ✓ | ✗ | 0.3400 | 0.3200 |
| STORM (2025) | CVPR25 | ✗ | ✗ | 0.1613 | 0.1981 |
| InfSplign (Ours) | | ✗ | ✗ | **0.3771** | **0.4172** |
| Improvement (Inference Time) | | | | +4.31% | +21.91% |
| Improvement (All) | | | | +3.71% | +9.72% |

Table 3: The impact of `InfSplign`'s loss terms $\mathcal{L}_{\text{spatial}}$ ($\mathcal{L}_s$), $\mathcal{L}_{\text{presence}}$ ($\mathcal{L}_p$) and $\mathcal{L}_{\text{balance}}$ ($\mathcal{L}_b$) on VISOR benchmark across two backbones.

| $\mathcal{L}_s$ | $\mathcal{L}_p$ | $\mathcal{L}_b$ | OA (%) | VISOR (%) | | | | | |
|---|---|---|---|---|---|---|---|---|---|
| | | | | uncond | cond | 1 | 2 | 3 | 4 |
| **Stable Diffusion 1.4** | | | | | | | | | |
| ✗ | ✗ | ✗ | 32.74 | 20.38 | 62.25 | 49.14 | 22.07 | 8.30 | 2.01 |
| ✗ | ✗ | ✓ | 33.47 | 21.10 | 63.03 | 50.34 | 23.26 | 8.79 | 1.99 |
| ✗ | ✓ | ✗ | 53.54 | 31.78 | 59.36 | 64.93 | 37.75 | 18.35 | 6.09 |
| ✗ | ✓ | ✓ | 54.35 | 32.46 | 59.73 | 65.93 | 38.72 | 18.84 | 6.37 |
| ✓ | ✗ | ✗ | 59.33 | 58.78 | 99.07 | 87.05 | 70.83 | 50.60 | 26.65 |
| ✓ | ✗ | ✓ | 60.09 | 59.55 | 99.09 | 87.54 | 71.83 | 51.69 | 27.13 |
| ✓ | ✓ | ✗ | 66.44 | 65.63 | 98.78 | 89.69 | 77.18 | 60.01 | 35.62 |
| ✓ | ✓ | ✓ | 67.36 | 66.54 | 98.79 | 90.48 | 77.79 | 61.37 | 36.54 |
| **Stable Diffusion 2.1** | | | | | | | | | |
| ✗ | ✗ | ✗ | 44.37 | 26.91 | 60.66 | 59.30 | 30.84 | 13.66 | 3.84 |
| ✗ | ✗ | ✓ | 43.32 | 27.21 | 62.82 | 60.56 | 31.31 | 13.18 | 3.79 |
| ✗ | ✓ | ✗ | 63.09 | 37.17 | 58.91 | 71.65 | 44.98 | 23.38 | 8.67 |
| ✗ | ✓ | ✓ | 62.85 | 37.69 | 59.97 | 72.98 | 45.92 | 23.41 | 8.45 |
| ✓ | ✗ | ✗ | 70.58 | 69.70 | 98.76 | 92.65 | 81.53 | 65.10 | 39.53 |
| ✓ | ✗ | ✓ | 71.12 | 70.18 | 98.69 | 92.90 | 82.20 | 65.76 | 39.86 |
| ✓ | ✓ | ✗ | 76.20 | 75.21 | 98.71 | 94.12 | 85.81 | 72.33 | 48.59 |
| ✓ | ✓ | ✓ | 77.28 | 76.26 | 98.68 | 94.65 | 86.66 | 73.48 | 50.23 |

**The effect of individual $\mathcal{L}_{\text{InfSplign}}$ terms.** Tab. 3 investigates the effect of the three loss terms across two backbones and demonstrates that all components contribute to improving baseline performance on both OA and VISOR metrics. $\mathcal{L}_{\text{spatial}}$ has the strongest impact on OA, and $\mathcal{L}_{\text{presence}}$ and $\mathcal{L}_{\text{balance}}$ come next in terms of impact in that order, incrementally improving the performance. As such, each and every component of the loss contributes to the overall strong margin beyond base SD v1.4 ($+34.62\%$). Combining $\mathcal{L}_{\text{spatial}}$ and $\mathcal{L}_{\text{presence}}$ seems to have a stronger impact than $\mathcal{L}_{\text{balance}}$ replacing the latter. This is expected as $\mathcal{L}_{\text{presence}}$ directly impacts OA, and thus the overall performance. Same trend applies to the other VISOR metrics as well as the stronger backbone SD v2.1.

**Qualitative Results.** Fig. 4 and Fig. 5 shed light on the actual generation capabilities of `InfSplign`. In Fig. 4 not only does the base SD fail to recognize the meaning of the spatial relationship from the prompt, but it also fails to generate both objects in unnatural object combinations. `InfSplign` significantly improves upon these limitations and synthesizes spatially-aligned images even in atypical object settings. In Fig. 5, we illustrate the generated samples by the base SD and competing inference-time baselines INITNO, CONFORM, and STORM. The results across both prompts corroborate that `InfSplign` honors spatial information the best among the competitors. Notably, the quality of generated images and subtle nuances therein can be further improved for all baselines with a stronger diffusion backbones such as SD v2.1 and SDXL. Additional examples can be found in Fig. C.

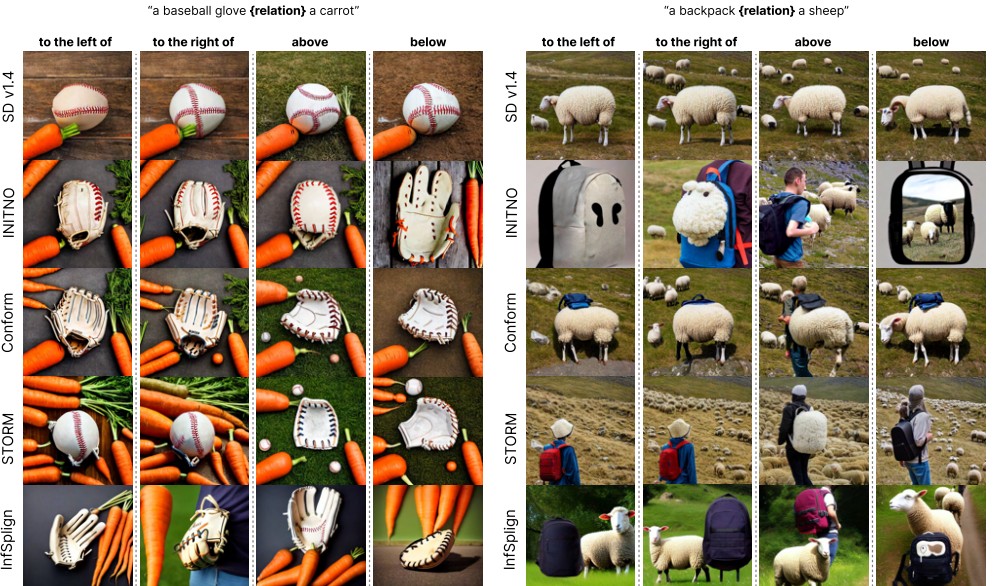

Figure 4: Qualitative comparison with SD across different VISOR prompts.

Figure 5: Comparison of spatial understanding across T2I diffusion models. `InfSplign` consistently aligns objects according to the target relation better than SD v1.4 (Rombach et al., 2022), INITNO (Guo et al., 2024), CONFORM (Meral et al., 2024) and STORM (Han et al., 2025).

## 5 CONCLUDING REMARKS

In this work, we tackle the challenge of spatial understanding in T2I diffusion models, a persistent limitation that undermines accurate compositional generation. We introduce `InfSplign`, a lightweight yet powerful inference-time method that improves spatial alignment and object preservation without requiring retraining or external supervision. Our approach applies a spatial loss during sampling, guided by attention maps to estimate object locations and variances. This not only enforces spatial relations specified in the prompt but also preserves object fidelity by regulating attention map variance. Crucially, `InfSplign` operates without modifying model weights or depending on auxiliary inputs such as bounding boxes or segmentation masks, making it fully modular and applicable to any diffusion model. Despite its simplicity, it surpasses fine-tuning-based methods and prior inference-time strategies aimed at spatial consistency, underscoring the promise of inference-time optimization for controllable and reliable generation.

**Limitations.** The base diffusion model often struggles to generate uncommon object combinations posing a challenge to any inference-time approach. For instance, VISOR includes a wide range of object pairs, many of which rarely co-occur in natural scenes. While this setup is valuable for testing generalization in spatial reasoning, it also makes the task more difficult, particularly when one or both objects fail to appear in the generated image. In such cases, `InfSplign` (and other inference-time baselines) cannot always effectively correct the spatial alignment, as the prerequisite object presence is impaired. As a result, object accuracy becomes the bottleneck in such scenarios.

**Large Language Models Usage.** We used ChatGPT (OpenAI) exclusively for minor sentence polishing, grammar improvements, and occasional formatting, as well as for assistance with resolving package dependencies.

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

# A  BENCHMARKS AND METRICS

## A.1  VISOR

VISOR (Gokhale et al., 2022) uses 80 object categories from the MS-COCO data set and defines unordered pairs of all objects, resulting in 6320 object combinations. Each object pair is matching with all 4 spatial relationships - "to the left of", "to the right of", "above" and "below". This gives us a total of 25280 spatial prompts. The VISOR dataset includes 6320 more prompts for each of the object pairs and the concept conjunction "and". Finally, it includes 80 more one-object prompts for all 80 COCO categories. In total, this results in 31680 text prompts. The VISOR benchmark defines the VISOR 1-4 scores, hence 4 images are generated per prompt. All VISOR images are generated with seed 42, ensuring that the images across prompts start from the same initial noise. We followed these guidelines in the evaluation of `InfSplign`.

The VISOR benchmark uses the OWL-ViT object detector to localize objects classes which are matched to the objects from the prompt. Then using a simple set of rules it predicts the spatial relationship and compares it with the ground truth relationship to determine if the image aligns with the spatial information in the prompt. The benchmark defines the following set of metrics: OA (object accuracy) indicating the number of images in which both objects were generated and detected, $\text{VISOR}_{\text{cond}}$ estimates the conditional probability of how many of the images adhere to the spatial relationship given that both objects are generated correctly, VISOR (or $\text{VISOR}_{\text{uncond}}$) gives a ratio of the number of images generated with a correct spatial relationship independent of the two objects being generated correctly. The $\text{VISOR}_1$ to $\text{VISOR}_N$ scores define the percentage of images out of N that have the correct spatial relationship between the objects present. All VISOR metrics are reported as a percentage.

## A.2  T2I-COMPBENCH

T2I-CompBench (Huang et al., 2023) is a for compositional T2I generation benchmark. It looks at attribute binding, object relationships and complex compositions. For our research, we focus on spatial alignment, hence focus on the 2D spatial relationships subcategory. The benchmark offers 1000 spatial prompts, but only 300 of them are used for testing since the authors present a fine-tuning approach and use the remaining 700 prompts for training. T2I-CompBench includes 7 spatial relationships. The first 4 relationships are directional and consistent with VISOR, while the last 3 are relative, requiring close proximity of objects: *"on the left of"*, *"on the right of"*, *"on the top of"*, *"on the bottom of"*, *"on the side of"*, *"next to"*, *"near"*. The objects are chosen from 3 categories: persons, animals and objects. In the benchmark the 10 images per prompt are generated with 10 different random seeds: 42...51, again ensuring that across prompts, the images start with the same reproducible initial noise.

T2I-CompBench uses the UniDet object detector to predict the location of each generated object. The centers of the objects are calculated from the detected bounding boxes. The metric estimates whether the spatial relationship is respected between both objects by comparing the centroids, measuring in which axis the objects are further apart, and the intersection-over-union (IoU), to avoid object overlap. To address the correctness of the 3 relative spatial relationships, the metric is calculated a weighted combination of confidence score for each of the two detected objects and the positional score.

## B   EXTENDED ABLATION STUDIES

To understand the effect of the interplay of hyperparameters on the spatial alignment score, a gridsearch for the parameters $\alpha$, $m$, $\lambda_s$, $\lambda_p$, $\lambda_b$ has been conducted. All experiments have been conducted on the $VISOR_{3160}$ subset, to keep experiments computationally feasible. The selected hyperparameters for SD v1.4 (Tab. 4) are: $\alpha = 1.5$, $m = 0.25$, $\lambda_s$=0.5, $\lambda_p$=1, $\lambda_b$=0.5. Alternatively the selected hyperparameters for SD v2.1 (Tab. 5) are: $\alpha = 1.5$, $m = 0.5$, $\lambda_s$=0.5, $\lambda_p$=1, $\lambda_b$=1.0.

Table 4: Grid search results for hyperparameter optimization for SD v1.4 on $VISOR_{3160}$

| $\alpha$ | $m$ | $\lambda_s$ | $\lambda_p$ | $\lambda_b$ | OA | $VISOR_{uncond}$ | $VISOR_{cond}$ | $VISOR_1$ | $VISOR_2$ | $VISOR_3$ | $VISOR_4$ |
|---|---|---|---|---|---|---|---|---|---|---|---|
| 0.5 | 0.25 | 0.5 | 0.5 | 0.5 | 57.08 | 53.27 | 93.32 | 81.87 | 63.99 | 44.84 | 22.37 |
| 0.5 | 0.25 | 0.5 | 0.5 | 1.0 | 58.35 | 54.45 | 93.32 | 83.07 | 65.29 | 45.63 | 23.83 |
| 0.5 | 0.25 | 0.5 | 1.0 | 0.5 | 58.26 | 51.97 | 89.20 | 81.23 | 62.53 | 42.72 | 21.39 |
| 0.5 | 0.25 | 0.5 | 1.0 | 1.0 | 58.92 | 52.46 | 89.04 | 81.71 | 63.45 | 43.20 | 21.49 |
| 0.5 | 0.25 | 1.0 | 0.5 | 0.5 | 63.01 | 61.67 | 97.87 | 87.22 | 72.85 | 55.89 | 30.73 |
| 0.5 | 0.25 | 1.0 | 0.5 | 1.0 | 63.54 | 62.26 | 97.97 | 87.79 | 73.80 | 55.89 | 31.55 |
| 0.5 | 0.25 | 1.0 | 1.0 | 0.5 | 63.68 | 61.56 | 96.67 | 86.93 | 73.35 | 54.65 | 31.30 |
| 0.5 | 0.25 | 1.0 | 1.0 | 1.0 | 63.90 | 61.95 | 96.94 | 87.53 | 73.54 | 55.19 | 31.52 |
| 0.5 | 0.50 | 0.5 | 0.5 | 0.5 | 58.68 | 55.27 | 94.19 | 83.23 | 66.36 | 46.46 | 25.03 |
| 0.5 | 0.50 | 0.5 | 0.5 | 1.0 | 59.04 | 55.97 | 94.80 | 84.21 | 66.74 | 47.75 | 25.16 |
| 0.5 | 0.50 | 0.5 | 1.0 | 0.5 | 59.03 | 53.75 | 91.06 | 82.06 | 64.72 | 45.60 | 22.63 |
| 0.5 | 0.50 | 0.5 | 1.0 | 1.0 | 59.98 | 54.88 | 91.51 | 83.86 | 65.73 | 46.11 | 23.83 |
| 0.5 | 0.50 | 1.0 | 0.5 | 0.5 | 64.72 | 63.65 | 98.34 | 88.61 | 75.48 | 56.93 | 33.58 |
| 0.5 | 0.50 | 1.0 | 0.5 | 1.0 | 65.08 | 64.00 | 98.35 | 89.37 | 75.44 | 58.07 | 33.13 |
| 0.5 | 0.50 | 1.0 | 1.0 | 0.5 | 64.42 | 62.96 | 97.73 | 87.22 | 73.89 | 57.63 | 33.10 |
| 0.5 | 0.50 | 1.0 | 1.0 | 1.0 | 65.12 | 63.58 | 97.63 | 87.63 | 74.56 | 58.29 | 33.83 |
| 0.5 | 0.75 | 0.5 | 0.5 | 0.5 | 59.50 | 56.66 | 95.23 | 84.46 | 67.60 | 48.73 | 25.85 |
| 0.5 | 0.75 | 0.5 | 0.5 | 1.0 | 60.32 | 57.62 | 95.53 | 85.29 | 69.27 | 49.08 | 26.84 |
| 0.5 | 0.75 | 0.5 | 1.0 | 0.5 | 59.88 | 55.57 | 92.80 | 83.29 | 66.23 | 47.88 | 24.87 |
| 0.5 | 0.75 | 0.5 | 1.0 | 1.0 | 60.83 | 56.80 | 93.38 | 84.40 | 68.13 | 48.64 | 26.04 |
| 0.5 | 0.75 | 1.0 | 0.5 | 0.5 | 65.23 | 64.45 | 98.81 | 88.96 | 76.80 | 58.73 | 33.32 |
| 0.5 | 0.75 | 1.0 | 0.5 | 1.0 | 65.92 | 65.19 | 98.90 | 89.62 | 77.37 | 59.46 | 34.30 |
| 0.5 | 0.75 | 1.0 | 1.0 | 0.5 | 65.77 | 64.58 | 98.20 | 88.96 | 75.92 | 58.80 | 34.65 |
| 0.5 | 0.75 | 1.0 | 1.0 | 1.0 | 65.32 | 64.12 | 98.16 | 88.32 | 75.35 | 58.32 | 34.49 |
| 1.0 | 0.25 | 0.5 | 0.5 | 0.5 | 64.28 | 63.20 | 98.31 | 87.91 | 74.72 | 57.15 | 33.01 |
| 1.0 | 0.25 | 0.5 | 0.5 | 1.0 | 64.57 | 63.42 | 98.22 | 88.73 | 74.97 | 57.31 | 32.66 |
| 1.0 | 0.25 | 0.5 | 1.0 | 0.5 | 64.72 | 63.24 | 97.70 | 87.91 | 74.68 | 57.06 | 33.29 |
| 1.0 | 0.25 | 0.5 | 1.0 | 1.0 | 65.02 | 63.51 | 97.68 | 88.04 | 74.18 | 57.72 | 34.11 |
| 1.0 | 0.25 | 1.0 | 0.5 | 0.5 | 65.97 | 65.44 | 99.21 | 90.73 | 78.01 | 59.81 | 33.23 |
| 1.0 | 0.25 | 1.0 | 0.5 | 1.0 | 66.46 | 65.94 | 99.23 | 90.98 | 78.13 | 60.85 | 33.80 |
| 1.0 | 0.25 | 1.0 | 1.0 | 0.5 | 67.74 | 67.24 | 99.26 | 90.63 | 78.96 | 62.18 | 37.18 |
| 1.0 | 0.25 | 1.0 | 1.0 | 1.0 | 67.90 | 67.40 | 99.25 | 91.33 | 79.30 | 62.09 | 36.87 |
| 1.0 | 0.50 | 0.5 | 0.5 | 0.5 | 65.64 | 64.87 | 98.82 | 89.65 | 76.52 | 59.24 | 34.05 |
| 1.0 | 0.50 | 0.5 | 0.5 | 1.0 | 66.38 | 65.62 | 98.86 | 90.25 | 77.69 | 59.65 | 34.87 |
| 1.0 | 0.50 | 0.5 | 1.0 | 0.5 | 66.35 | 65.30 | 98.41 | 89.15 | 76.71 | 59.78 | 35.57 |
| 1.0 | 0.50 | 0.5 | 1.0 | 1.0 | 66.37 | 65.25 | 98.32 | 89.75 | 76.01 | 59.65 | 35.60 |
| 1.0 | 0.50 | 1.0 | 0.5 | 0.5 | 63.67 | 63.26 | 99.35 | 90.03 | 75.76 | 56.39 | 30.85 |
| 1.0 | 0.50 | 1.0 | 0.5 | 1.0 | 63.70 | 63.20 | 99.22 | 90.19 | 75.70 | 56.49 | 30.41 |
| 1.0 | 0.50 | 1.0 | 1.0 | 0.5 | 65.88 | 65.37 | 99.23 | 91.01 | 77.88 | 59.18 | 33.42 |
| 1.0 | 0.50 | 1.0 | 1.0 | 1.0 | 66.60 | 66.17 | 99.36 | 91.39 | 78.83 | 59.75 | 34.72 |
| 1.0 | 0.75 | 0.5 | 0.5 | 0.5 | 66.18 | 65.51 | 99.00 | 90.29 | 77.79 | 60.22 | 33.77 |
| 1.0 | 0.75 | 0.5 | 0.5 | 1.0 | 66.14 | 65.51 | 99.04 | 90.85 | 78.58 | 60.06 | 32.53 |
| 1.0 | 0.75 | 0.5 | 1.0 | 0.5 | 66.44 | 65.56 | 98.68 | 90.03 | 77.15 | 60.67 | 34.40 |
| 1.0 | 0.75 | 0.5 | 1.0 | 1.0 | 67.62 | 66.76 | 98.73 | 90.29 | 78.26 | 61.99 | 36.49 |
| 1.0 | 0.75 | 1.0 | 0.5 | 0.5 | 61.05 | 60.64 | 99.33 | 88.29 | 73.29 | 53.01 | 27.98 |
| 1.0 | 0.75 | 1.0 | 0.5 | 1.0 | 61.78 | 61.41 | 99.40 | 90.10 | 74.43 | 53.10 | 28.01 |
| 1.0 | 0.75 | 1.0 | 1.0 | 0.5 | 63.39 | 63.02 | 99.43 | 90.41 | 75.67 | 55.85 | 30.16 |
| 1.0 | 0.75 | 1.0 | 1.0 | 1.0 | 63.70 | 63.30 | 99.37 | 90.63 | 76.20 | 55.98 | 30.38 |
| 1.5 | 0.25 | 0.5 | 0.5 | 0.5 | 66.46 | 65.95 | 99.23 | 90.06 | 78.17 | 60.70 | 34.87 |
| 1.5 | 0.25 | 0.5 | 0.5 | 1.0 | 66.70 | 66.08 | 99.06 | 90.41 | 79.30 | 60.38 | 34.21 |
| 1.5 | 0.25 | 0.5 | 1.0 | 0.5 | 67.96 | 67.26 | 98.96 | 90.38 | 78.54 | 62.79 | 37.31 |
| 1.5 | 0.25 | 0.5 | 1.0 | 1.0 | 67.94 | 67.20 | 98.92 | 90.41 | 78.92 | 61.77 | 37.69 |
| 1.5 | 0.25 | 1.0 | 0.5 | 0.5 | 61.61 | 61.16 | 99.26 | 89.49 | 73.54 | 53.51 | 28.07 |
| 1.5 | 0.25 | 1.0 | 0.5 | 1.0 | 62.38 | 61.96 | 99.33 | 89.40 | 75.76 | 54.53 | 28.17 |
| 1.5 | 0.25 | 1.0 | 1.0 | 0.5 | 64.11 | 63.73 | 99.42 | 90.29 | 76.71 | 57.34 | 30.60 |
| 1.5 | 0.25 | 1.0 | 1.0 | 1.0 | 64.80 | 64.35 | 99.30 | 90.89 | 77.44 | 57.56 | 31.52 |
| 1.5 | 0.50 | 0.5 | 0.5 | 0.5 | 64.98 | 64.53 | 99.32 | 90.19 | 77.15 | 58.39 | 32.41 |
| 1.5 | 0.50 | 0.5 | 0.5 | 1.0 | 65.55 | 65.08 | 99.29 | 90.98 | 77.72 | 59.30 | 32.31 |
| 1.5 | 0.50 | 0.5 | 1.0 | 0.5 | 67.56 | 66.92 | 99.05 | 91.30 | 78.89 | 60.92 | 36.55 |
| 1.5 | 0.50 | 0.5 | 1.0 | 1.0 | 67.56 | 67.08 | 99.30 | 91.49 | 79.56 | 61.77 | 35.51 |
| 1.5 | 0.50 | 1.0 | 0.5 | 0.5 | 53.77 | 53.46 | 99.43 | 84.78 | 65.29 | 43.29 | 20.48 |
| 1.5 | 0.50 | 1.0 | 0.5 | 1.0 | 54.08 | 53.72 | 99.33 | 85.32 | 66.14 | 43.54 | 19.87 |
| 1.5 | 0.50 | 1.0 | 1.0 | 0.5 | 56.95 | 56.65 | 99.47 | 87.44 | 68.86 | 47.28 | 23.01 |
| 1.5 | 0.50 | 1.0 | 1.0 | 1.0 | 57.29 | 56.95 | 99.42 | 87.47 | 69.56 | 46.90 | 23.89 |
| 1.5 | 0.75 | 0.5 | 0.5 | 0.5 | 62.89 | 62.43 | 99.27 | 89.65 | 74.78 | 55.92 | 29.37 |
| 1.5 | 0.75 | 0.5 | 0.5 | 1.0 | 63.20 | 62.82 | 99.39 | 90.16 | 75.73 | 55.67 | 29.72 |
| 1.5 | 0.75 | 0.5 | 1.0 | 0.5 | 65.61 | 65.13 | 99.26 | 90.38 | 77.41 | 58.83 | 33.89 |
| 1.5 | 0.75 | 0.5 | 1.0 | 1.0 | 65.51 | 65.06 | 99.31 | 90.29 | 77.88 | 59.18 | 32.88 |
| 1.5 | 0.75 | 1.0 | 0.5 | 0.5 | 47.70 | 47.40 | 99.37 | 80.85 | 59.02 | 35.38 | 14.34 |
| 1.5 | 0.75 | 1.0 | 0.5 | 1.0 | 47.66 | 47.38 | 99.42 | 80.98 | 58.10 | 35.67 | 14.78 |
| 1.5 | 0.75 | 1.0 | 1.0 | 0.5 | 49.64 | 49.38 | 99.49 | 83.20 | 60.76 | 37.60 | 15.98 |
| 1.5 | 0.75 | 1.0 | 1.0 | 1.0 | 49.68 | 49.41 | 99.44 | 82.98 | 61.01 | 37.88 | 15.76 |

Table 5: Grid search results for hyperparameter optimization for SDv2.1 on $VISOR_{3160}$

| $\alpha$ | $m$ | $\lambda_s$ ($\mathcal{L}_{\text{spatial}}$) | $\lambda_p$ ($\mathcal{L}_{\text{presence}}$) | $\lambda_b$ (($\mathcal{L}_{\text{balance}}$)) | OA | $VISOR_{uncond}$ | $VISOR_{cond}$ | $VISOR_1$ | $VISOR_2$ | $VISOR_3$ | $VISOR_4$ |
|---|---|---|---|---|---|---|---|---|---|---|---|
| 0.5 | 0.25 | 0.5 | 0.5 | 0.5 | 65.49 | 61.23 | 93.49 | 86.90 | 71.68 | 54.56 | 31.77 |
| 0.5 | 0.25 | 0.5 | 0.5 | 1.0 | 65.86 | 61.88 | 93.96 | 87.91 | 73.07 | 54.94 | 31.61 |
| 0.5 | 0.25 | 0.5 | 1.0 | 0.5 | 67.84 | 61.45 | 90.58 | 87.15 | 72.37 | 54.59 | 31.68 |
| 0.5 | 0.25 | 0.5 | 1.0 | 1.0 | 67.91 | 62.07 | 91.40 | 88.20 | 72.47 | 54.81 | 32.82 |
| 0.5 | 0.25 | 1.0 | 0.5 | 0.5 | 71.26 | 69.51 | 97.55 | 91.65 | 80.13 | 64.11 | 42.15 |
| 0.5 | 0.25 | 1.0 | 0.5 | 1.0 | 71.17 | 69.53 | 97.69 | 91.52 | 80.25 | 64.53 | 41.80 |
| 0.5 | 0.25 | 1.0 | 1.0 | 0.5 | 73.71 | 71.35 | 96.80 | 92.22 | 81.23 | 67.22 | 44.75 |
| 0.5 | 0.25 | 1.0 | 1.0 | 1.0 | 73.83 | 71.57 | 96.95 | 92.47 | 81.36 | 66.93 | 45.54 |
| 0.5 | 0.50 | 0.5 | 0.5 | 0.5 | 66.61 | 62.99 | 94.56 | 87.75 | 73.45 | 56.68 | 34.08 |
| 0.5 | 0.50 | 0.5 | 0.5 | 1.0 | 66.52 | 63.16 | 94.95 | 88.70 | 74.43 | 56.49 | 33.01 |
| 0.5 | 0.50 | 0.5 | 1.0 | 0.5 | 68.88 | 63.52 | 92.21 | 88.83 | 74.11 | 56.52 | 34.62 |
| 0.5 | 0.50 | 0.5 | 1.0 | 1.0 | 69.24 | 64.45 | 93.08 | 89.43 | 75.41 | 57.85 | 35.10 |
| 0.5 | 0.50 | 1.0 | 0.5 | 0.5 | 72.63 | 71.15 | 97.96 | 92.37 | 81.61 | 66.77 | 43.83 |
| 0.5 | 0.50 | 1.0 | 0.5 | 1.0 | 71.83 | 70.36 | 97.95 | 92.25 | 80.98 | 65.19 | 43.01 |
| 0.5 | 0.50 | 1.0 | 1.0 | 0.5 | 74.55 | 72.63 | 97.42 | 92.91 | 82.63 | 68.29 | 46.68 |
| 0.5 | 0.50 | 1.0 | 1.0 | 1.0 | 75.11 | 73.26 | 97.54 | 93.73 | 82.98 | 69.24 | 47.09 |
| 0.5 | 0.75 | 0.5 | 0.5 | 0.5 | 67.81 | 64.92 | 95.74 | 89.37 | 75.25 | 58.73 | 36.33 |
| 0.5 | 0.75 | 0.5 | 0.5 | 1.0 | 68.23 | 65.21 | 95.57 | 89.56 | 75.76 | 58.99 | 36.52 |
| 0.5 | 0.75 | 0.5 | 1.0 | 0.5 | 70.20 | 65.89 | 93.87 | 89.40 | 76.84 | 59.97 | 37.37 |
| 0.5 | 0.75 | 0.5 | 1.0 | 1.0 | 70.43 | 66.55 | 94.50 | 90.63 | 77.44 | 60.63 | 37.50 |
| 0.5 | 0.75 | 1.0 | 0.5 | 0.5 | 73.88 | 72.51 | 98.15 | 92.53 | 82.82 | 68.48 | 46.20 |
| 0.5 | 0.75 | 1.0 | 0.5 | 1.0 | 73.62 | 72.29 | 98.20 | 93.20 | 82.72 | 68.29 | 44.97 |
| 0.5 | 0.75 | 1.0 | 1.0 | 0.5 | 75.65 | 74.18 | 98.06 | 93.13 | 83.96 | 70.98 | 48.64 |
| 0.5 | 0.75 | 1.0 | 1.0 | 1.0 | 74.92 | 73.36 | 97.92 | 93.17 | 83.26 | 69.27 | 47.75 |
| 1.0 | 0.25 | 0.5 | 0.5 | 0.5 | 72.04 | 70.63 | 98.05 | 92.06 | 81.08 | 66.36 | 43.04 |
| 1.0 | 0.25 | 0.5 | 0.5 | 1.0 | 72.10 | 70.74 | 98.12 | 91.96 | 82.03 | 66.42 | 42.56 |
| 1.0 | 0.25 | 0.5 | 1.0 | 0.5 | 74.32 | 72.41 | 97.44 | 93.04 | 82.18 | 68.45 | 45.98 |
| 1.0 | 0.25 | 0.5 | 1.0 | 1.0 | 74.79 | 72.94 | 97.51 | 93.42 | 82.85 | 68.70 | 46.77 |
| 1.0 | 0.25 | 1.0 | 0.5 | 0.5 | 75.40 | 74.51 | 98.83 | 94.08 | 85.03 | 71.27 | 47.66 |
| 1.0 | 0.25 | 1.0 | 0.5 | 1.0 | 75.22 | 74.29 | 98.76 | 93.99 | 85.70 | 71.11 | 46.36 |
| 1.0 | 0.25 | 1.0 | 1.0 | 0.5 | 76.98 | 76.02 | 98.76 | 94.68 | 85.63 | 73.20 | 50.57 |
| 1.0 | 0.25 | 1.0 | 1.0 | 1.0 | 77.89 | 76.95 | 98.80 | 95.16 | 87.18 | 75.35 | 50.13 |
| 1.0 | 0.50 | 0.5 | 0.5 | 0.5 | 74.28 | 72.97 | 98.23 | 93.07 | 83.29 | 69.91 | 45.60 |
| 1.0 | 0.50 | 0.5 | 0.5 | 1.0 | 73.97 | 72.81 | 98.43 | 93.70 | 83.54 | 68.67 | 45.32 |
| 1.0 | 0.50 | 0.5 | 1.0 | 0.5 | 75.69 | 74.23 | 98.08 | 93.58 | 83.83 | 71.04 | 48.48 |
| 1.0 | 0.50 | 0.5 | 1.0 | 1.0 | 76.16 | 74.70 | 98.09 | 93.96 | 85.35 | 70.38 | 49.11 |
| 1.0 | 0.50 | 1.0 | 0.5 | 0.5 | 74.34 | 73.68 | 99.11 | 94.11 | 84.94 | 70.51 | 45.16 |
| 1.0 | 0.50 | 1.0 | 0.5 | 1.0 | 74.67 | 73.90 | 98.97 | 93.99 | 85.10 | 70.76 | 45.76 |
| 1.0 | 0.50 | 1.0 | 1.0 | 0.5 | 76.89 | 75.97 | 98.81 | 95.06 | 86.77 | 73.54 | 48.51 |
| 1.0 | 0.50 | 1.0 | 1.0 | 1.0 | 76.69 | 75.84 | 98.90 | 95.22 | 86.61 | 73.17 | 48.35 |
| 1.0 | 0.75 | 0.5 | 0.5 | 0.5 | 74.59 | 73.53 | 98.58 | 93.64 | 84.02 | 69.84 | 46.61 |
| 1.0 | 0.75 | 0.5 | 0.5 | 1.0 | 75.09 | 73.99 | 98.54 | 93.99 | 84.72 | 70.95 | 46.30 |
| 1.0 | 0.75 | 0.5 | 1.0 | 0.5 | 76.62 | 75.37 | 98.37 | 93.61 | 85.63 | 72.94 | 49.30 |
| 1.0 | 0.75 | 0.5 | 1.0 | 1.0 | 77.15 | 76.04 | 98.56 | 94.62 | 86.61 | 73.10 | 49.84 |
| 1.0 | 0.75 | 1.0 | 0.5 | 0.5 | 72.90 | 72.15 | 98.97 | 93.96 | 84.27 | 68.13 | 42.25 |
| 1.0 | 0.75 | 1.0 | 0.5 | 1.0 | 73.10 | 72.40 | 99.04 | 94.24 | 84.18 | 69.43 | 41.74 |
| 1.0 | 0.75 | 1.0 | 1.0 | 0.5 | 74.98 | 74.11 | 98.85 | 93.96 | 86.08 | 71.17 | 45.25 |
| 1.0 | 0.75 | 1.0 | 1.0 | 1.0 | 75.25 | 74.42 | 98.89 | 94.84 | 85.51 | 71.33 | 45.98 |
| 1.5 | 0.25 | 0.5 | 0.5 | 0.5 | 75.06 | 74.13 | 98.76 | 93.48 | 84.62 | 71.11 | 47.31 |
| 1.5 | 0.25 | 0.5 | 0.5 | 1.0 | 75.11 | 74.11 | 98.66 | 94.05 | 85.41 | 70.57 | 46.39 |
| 1.5 | 0.25 | 0.5 | 1.0 | 0.5 | 76.69 | 75.49 | 98.44 | 93.86 | 85.57 | 72.75 | 49.78 |
| 1.5 | 0.25 | 0.5 | 1.0 | 1.0 | 76.84 | 75.77 | 98.61 | 93.83 | 85.51 | 72.94 | 50.79 |
| 1.5 | 0.25 | 1.0 | 0.5 | 0.5 | 72.24 | 71.62 | 99.15 | 94.02 | 83.70 | 67.53 | 41.23 |
| 1.5 | 0.25 | 1.0 | 0.5 | 1.0 | 72.22 | 71.46 | 98.95 | 93.35 | 83.67 | 67.85 | 40.95 |
| 1.5 | 0.25 | 1.0 | 1.0 | 0.5 | 74.78 | 74.05 | 99.03 | 94.91 | 85.54 | 71.17 | 44.59 |
| 1.5 | 0.25 | 1.0 | 1.0 | 1.0 | 74.80 | 73.99 | 98.91 | 94.49 | 85.73 | 70.89 | 44.84 |
| 1.5 | 0.50 | 0.5 | 0.5 | 0.5 | 75.01 | 74.12 | 98.82 | 93.89 | 84.97 | 70.98 | 46.65 |
| 1.5 | 0.50 | 0.5 | 0.5 | 1.0 | 75.06 | 74.19 | 98.83 | 94.08 | 85.19 | 71.04 | 46.42 |
| 1.5 | 0.50 | 0.5 | 1.0 | 0.5 | 77.36 | 76.43 | 98.80 | 94.62 | 86.36 | 73.89 | 50.85 |
| 1.5 | 0.50 | 0.5 | 1.0 | 1.0 | 77.98 | 77.10 | 98.86 | 95.00 | 87.18 | 74.40 | 51.80 |
| 1.5 | 0.50 | 1.0 | 0.5 | 0.5 | 67.41 | 66.77 | 99.05 | 92.18 | 80.00 | 61.74 | 33.17 |
| 1.5 | 0.50 | 1.0 | 0.5 | 1.0 | 66.80 | 66.11 | 98.97 | 91.42 | 79.34 | 60.35 | 33.32 |
| 1.5 | 0.50 | 1.0 | 1.0 | 0.5 | 69.17 | 68.55 | 99.11 | 93.42 | 81.23 | 63.39 | 36.17 |
| 1.5 | 0.50 | 1.0 | 1.0 | 1.0 | 69.22 | 68.58 | 99.09 | 93.17 | 81.11 | 63.45 | 36.61 |
| 1.5 | 0.75 | 0.5 | 0.5 | 0.5 | 74.26 | 73.52 | 99.01 | 93.86 | 85.10 | 69.78 | 45.35 |
| 1.5 | 0.75 | 0.5 | 0.5 | 1.0 | 74.72 | 73.90 | 98.91 | 94.72 | 84.91 | 70.60 | 45.38 |
| 1.5 | 0.75 | 0.5 | 1.0 | 0.5 | 76.80 | 76.04 | 99.01 | 94.34 | 87.03 | 73.73 | 49.08 |
| 1.5 | 0.75 | 0.5 | 1.0 | 1.0 | 76.76 | 75.89 | 98.87 | 95.00 | 86.11 | 73.32 | 49.11 |
| 1.5 | 0.75 | 1.0 | 0.5 | 0.5 | 61.66 | 61.12 | 99.12 | 89.05 | 74.37 | 53.13 | 27.91 |
| 1.5 | 0.75 | 1.0 | 0.5 | 1.0 | 61.54 | 60.97 | 99.07 | 90.19 | 74.27 | 52.85 | 26.55 |
| 1.5 | 0.75 | 1.0 | 1.0 | 0.5 | 64.73 | 64.15 | 99.10 | 91.36 | 77.56 | 57.41 | 30.25 |
| 1.5 | 0.75 | 1.0 | 1.0 | 1.0 | 64.65 | 63.98 | 98.96 | 91.01 | 76.84 | 57.56 | 30.51 |

## C   EXTRA QUALITATIVE RESULTS

Fig. 4 showcases the better generation power of `InfSplign` compared to the Stable Diffusion baseline. The spatial losses successfully overcome the limitations in the baseline - incorrect spatial placement and single object generation. The base model generated results where objects vary in spatial location, which implies the ignorance of the base SD to the spatial relationship mentioned in the prompt. In Fig. 4, examples with spatial relations *"left of"* and *"right of"* generate two misaligned objects, whereas with *"above"* and *"below"* the model struggles to generate both objects in one image. We attribute this mostly to the unnatural combination of the objects in the prompt: e.g. "a bench **above** a cake" (Chefer et al., 2023). Hence, the best that SD can do is to only generate object combinations that it has seen during training - it is more likely to produce a "cat" together with a "motorcycle" than a "cake" and a "bench" in one image. The missing object problem can also be explained with some of the insights discussed in A&E (Chefer et al., 2023), namely that it can be suppressed, mixed with the other object, entangled in the representation of the other object or subtly blended in the image.

`InfSplign` successfully addresses the constraints that the base SD faces by introducing well-crafted spatial losses which produce a meaningful signal used to guide the underlying diffusion model through the denoising process. Our method successfully interpreted the spatial information and generated both object at locations in accordance with the spatial relationship given in the prompt. In the rare object combination case, "a bench **above** a cake", `InfSplign` successfully disentangled the concept of the "bench" from the attention map and cleverly figures out that the bench object cannot realistically be placed on top of a cake, so it generates it as a cake topper above the cake.

To further exhibit the power of `InfSplign` to spatially align objects in the image, Fig. 6 through Fig. 15 showcase qualitative comparisons with the relevant competitors (SD v1.4 (Rombach et al., 2022), INITNO (Guo et al., 2024), CONFORM (Meral et al., 2024) and STORM (Han et al., 2025)). Note that `InfSplign` qualitatively proves the gains showcased in the earlier mentioned quantitative analysis in Tab. 1 and Tab. 2.

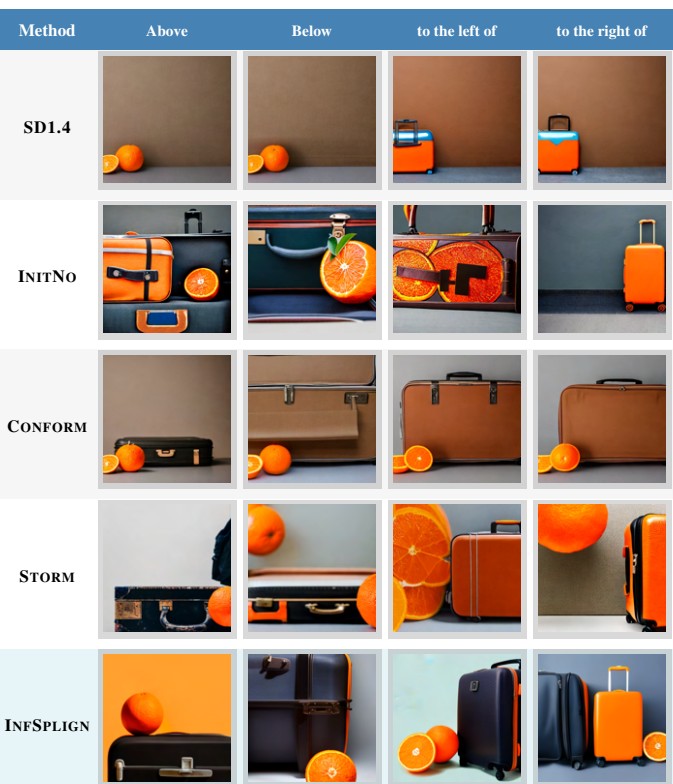

Figure 6: An Orange {relation} a Suitcase.

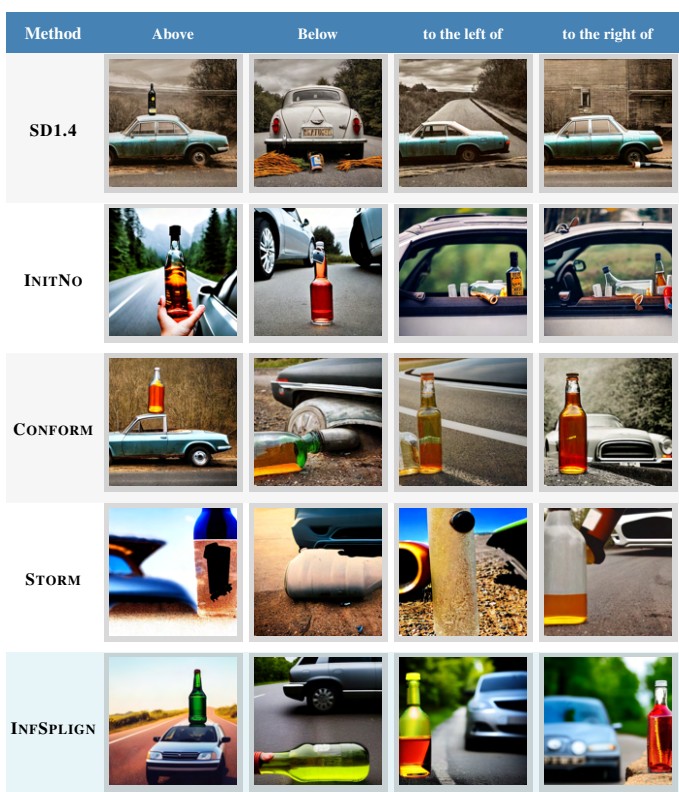

Figure 7: A Bottle {relation} a Car.

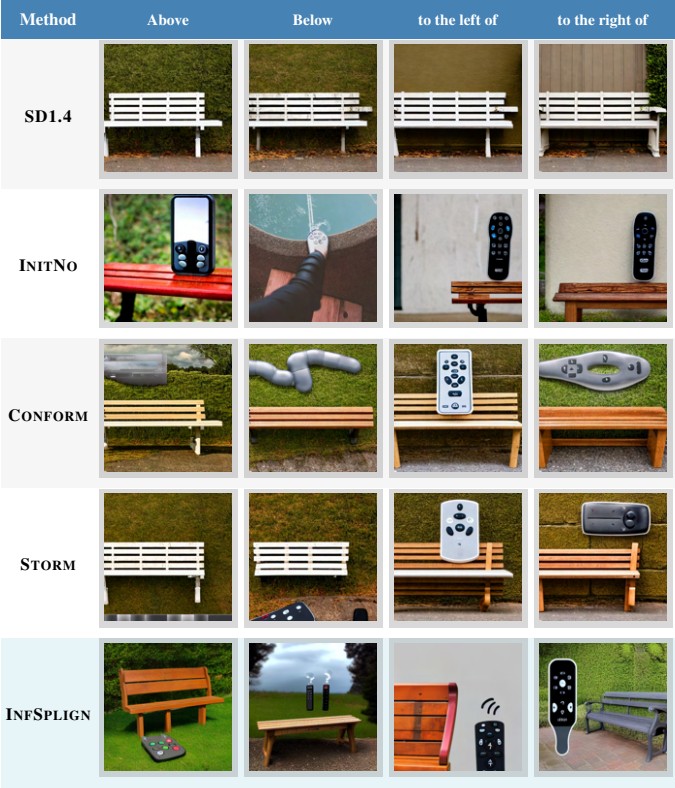

Figure 8: A Bench {relation} a Remote.

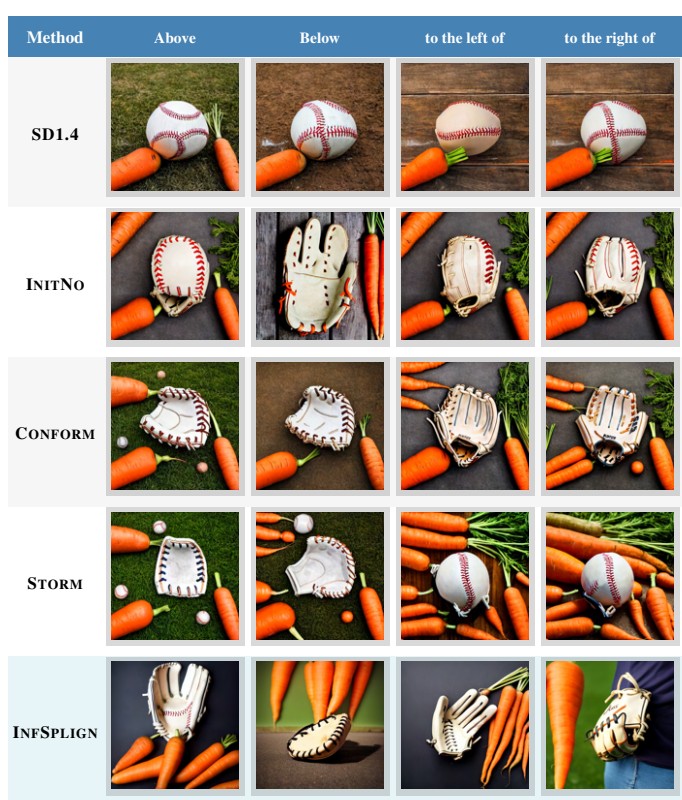

Figure 9: A Baseball Glove {relation} a Carrot.

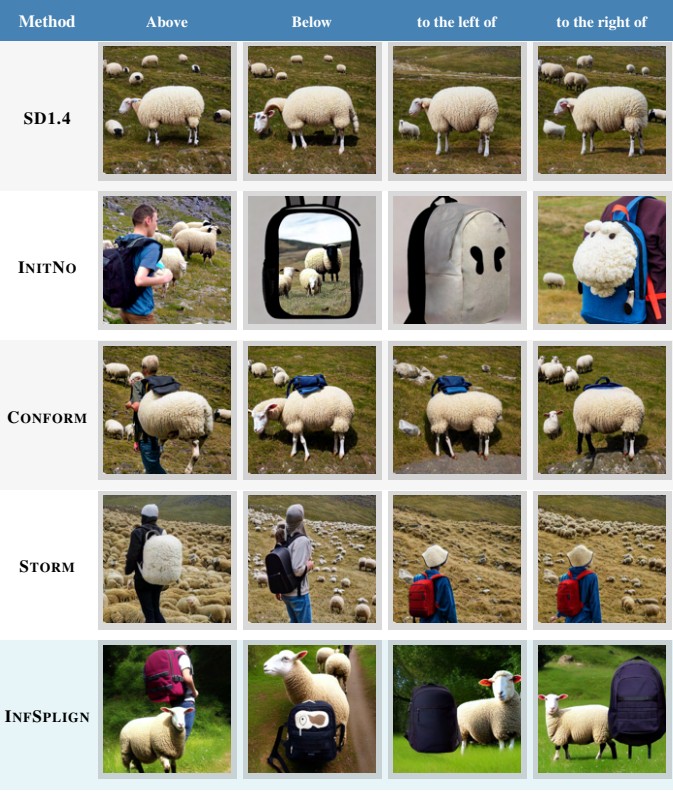

Figure 10: A Backpack {relation} a Sheep.

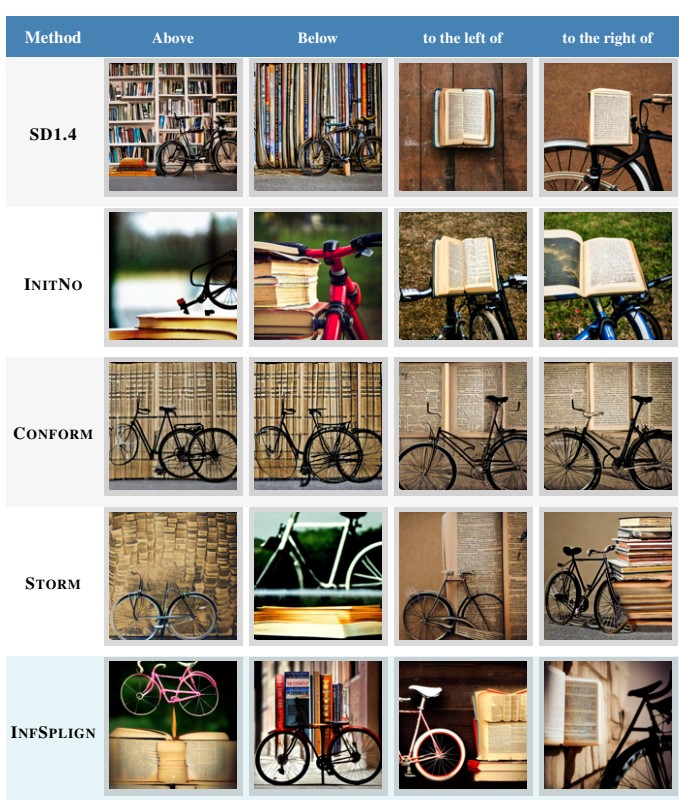

Figure 11: A Bicycle {relation} a Book.

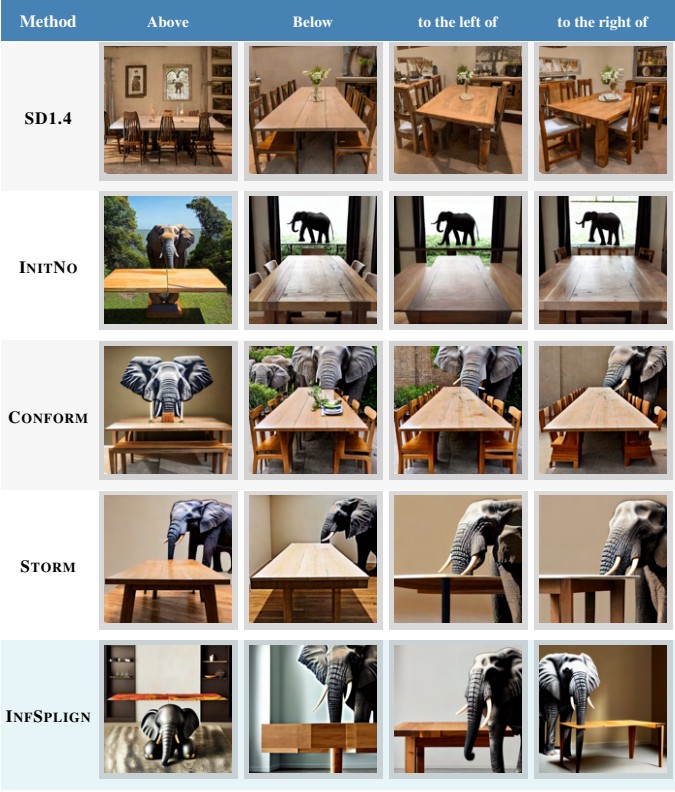

Figure 12: A Dining Table {relation} an Elephant.

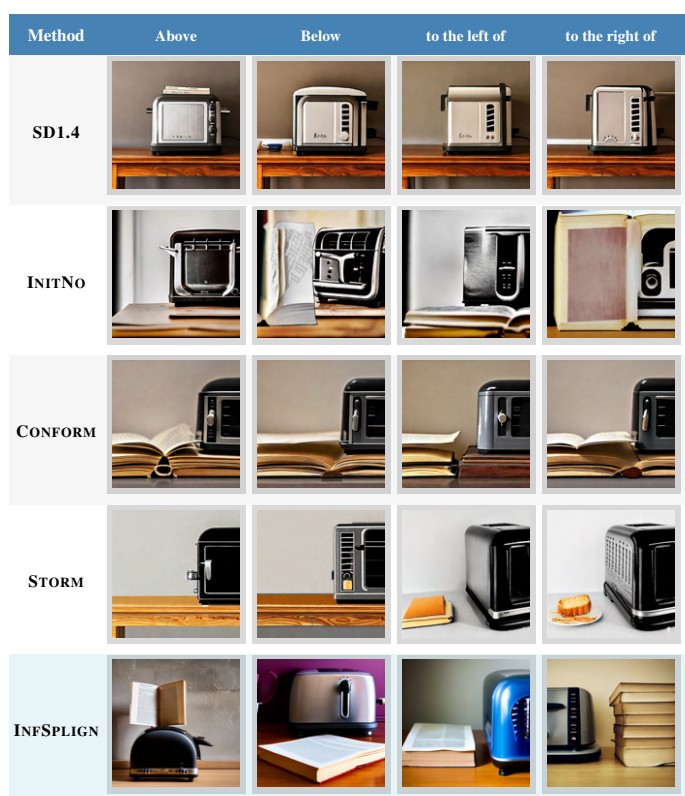

Figure 13: A Book {relation} a Toaster.

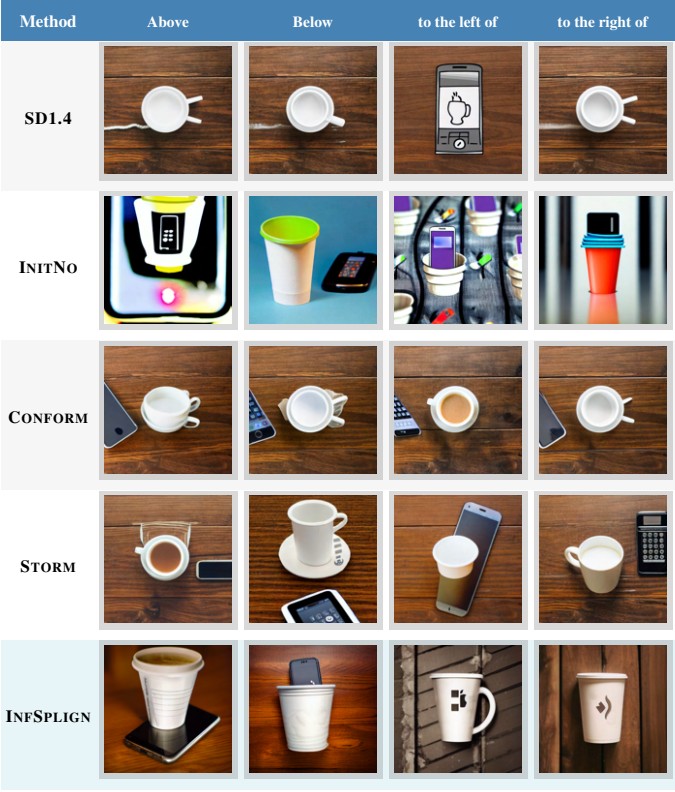

Figure 14: A Cup {relation} a Cell Phone.

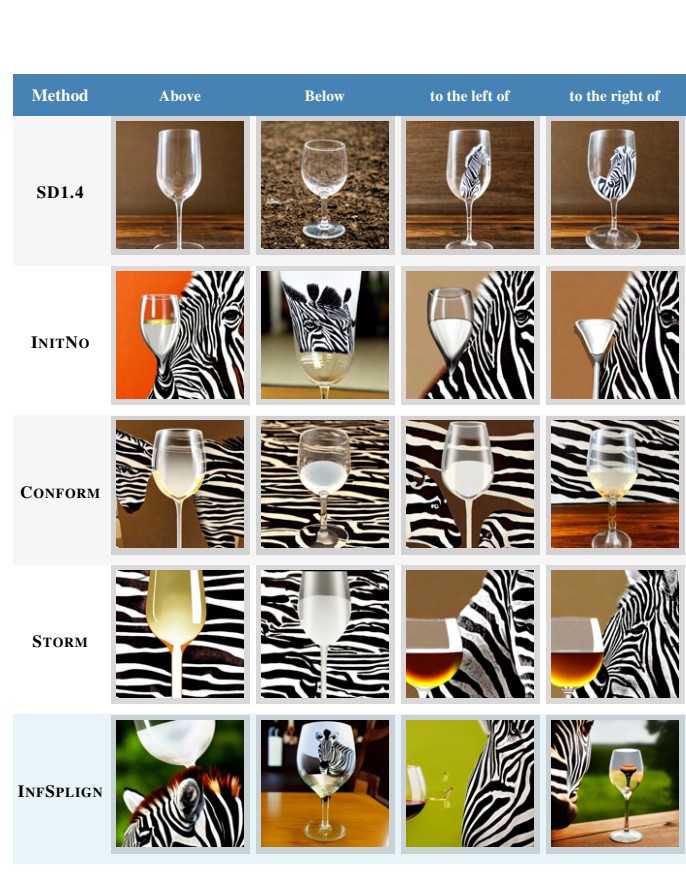

Figure 15: A Wine Glass {relation} a Zebra.

