# OpenReview forum: "InfSplign: Inference-Time Spatial Alignment of Text-to-Image Diffusion Models"
_ICLR.cc/2026/Conference — ICLR 2026 Conference Withdrawn Submission_

### Official Review · Reviewer_R376 · 2025-10-28

**Soundness:** 3
**Presentation:** 2
**Contribution:** 3
**Rating:** 4
**Confidence:** 3

**Summary:**

This paper proposes InfSplign, a training-free, inference-time spatial alignment method for text-to-image diffusion models (Stable Diffusion). The approach computes cross-attention maps for each object token at multiple decoder layers, estimates centroids and variances, and applies a composite spatial loss during sampling. Extensive experiments on VISOR and T2I-CompBench demonstrate the proposed method.

**Strengths:**

- The formulation is intuitive and mathematically grounded in attention-space statistics.
- Outperform in both inference-time and fine-tuning methods on major spatial reasoning benchmarks.
- Ablation results demonstrate that all three loss components contribute meaningfully.

**Weaknesses:**

- The paper lacks a convincing justification that cross-attention variance is a valid proxy for object presence and representation balance; while the method computes centroids and variances from decoder cross-attention, it does not establish (theoretically or empirically) that lower variance reliably implies preserved objects or balanced representations, which calls for direct evidence.
- The paper lacks a clean isolation of spatial-alignment gains relative to STORM; results are reported only for the combined objective. so it remains unclear whether improvements stem from genuine spatial reasoning ($L_{spatial}$) or from easier benefits due to improved object visibility or balance ($L_{precense}, L_{balance}$). A fair comparison requires reporting an $L_{spatial}$ only variant against STORM and fine-grained ablations with presence and balance removed.
- The runtime comparison is missing: how does the end-to-end inference time in vanilla Stable Diffusion compare to that of the proposed method?

**Questions:**

Does the method extend to transformer-based diffusion backbones (e.g, DiT, FLUX)? If so, what modifications are required to the attention hooks to make it work on these architectures?

---

### Official Review · Reviewer_4aiC · 2025-10-29

**Soundness:** 3
**Presentation:** 3
**Contribution:** 2
**Rating:** 2
**Confidence:** 5

**Summary:**

The paper introduces InfSplign, a training-free, inference-time guidance method to improce the spatial alignment of the text-to-image diffusion models. The core idea is to apply a compound loss over the features extracted from the attention maps of the UNet during inference time to nudge the noisy latent towards correct spatial alignment. The method is simple, intuitive, and demostrates impressive state-of-the art performance on the VISOR and T2I-CompBench benchmarks, even outperforming fine-tuning based approaches.

However, the works' significant contribution is undermined by several critical weaknesses, The primary spatial loss is based on handcrafted, non-generalized set of rules that appear overfitted to the specific relations in the VISOR benchmark. Furthermore, The object presence and balance losses rely on heuristics that may fail in common compositional scenarios involving nested objects or objects of separate scales.

**Strengths:**

- The method is simple, effective and plug-and play. Since the method doesn't require any retraining, it can be applied to Unet-based models.
- Spatial placement (centroid margin), object dropping (variance minimization), and overshadowing (variance parity) map cleanly onto observed issues.
- VISOR and T2I-CompBench numbers are consistently higher than both inference-time and fine-tuning-based baselines.

**Weaknesses:**

- The spatial loss encodes a fixed set of relations (left/right/above/below/near) via axis-wise centroid differences with a margin. This is likely tuned to VISOR and does not cover richer or contextual relations (between, around, on, behind/in-front-of).
- The method assumes the caption can be parsed cleanly into ⟨A,R,B⟩ and only enforces a single binary relation at a time. The paper does not study multi-object scenes with multiple simultaneous constraints, where losses could conflict.
- The control relies on explicit, handcrafted geometry and a handful of tuned hyperparameters (α, m, λ’s), selected via grid search on a VISOR subset. This risks overfitting the control surface to that distribution.

**Questions:**

- Your spatial loss in Eq. 6 is defined for a fixed set of five relations. How would your method handle more complex, non-axial relations like "a car between two trees" or "a fence surrounding a house"? Furthermore, the T2I-CompBench includes relations like "on the side of," which is not in your list; how is this handled by your implementation? Does it default to the "near" condition, and if so, how does this affect the precision of the guidance?

- The representation balance loss $\mathcal{L}_{balance}$ penalizes differences in attention variance, implicitly assuming objects should have a similar "footprint" in the attention space. How does this loss behave in scenarios with intended scale differences or nesting, such as "a man holding his child" or "a sticker on a laptop"? Would this loss not provide a counterproductive signal in these common cases by trying to equalize the variance?

- Your method uses centroids as a proxy for object location. STORM, another high-performing inference-time method, uses Optimal Transport to reshape the entire attention distribution. Could you please provide a conceptual comparison between your versus optimal transport? What are the fundamental advantages and disadvantages of each, beyond final benchmark scores?

---

### Official Review · Reviewer_RCVb · 2025-10-30

**Soundness:** 2
**Presentation:** 2
**Contribution:** 2
**Rating:** 2
**Confidence:** 4

**Summary:**

This paper proposes a training-free, plug-and-play method to improve spatial alignment of T2I diffusion models. The approach involves adjusting the noise during each denoising step using a combination of three loss functions. The authors demonstrate the method's efficacy on U-Net-based diffusion models, specifically SD1.4 and 2.1.

**Strengths:**

- The proposed method is training-free and can be integrated into existing models as a plug-and-play module, which can be a good practical method for U-Net based T2I diffusion models.
- The paper includes extensive ablation studies that thoroughly investigate the contributions of different components, particularly the loss hyperparameters.

**Weaknesses:**

- The experimental validation is limited to relatively older U-Net architectures (SD1.4, SD2.1). It is unclear whether the spatial alignment issues addressed persist in more recent, state-of-the-art models, and whether the proposed method remains effective on them.
- The method's design is heavily focused on the U-Net architecture, which has shifted to transformer-based backbones such as DiT and MMDiT (e.g., in SD3). These newer architectures employ different attention mechanisms (e.g., joint attention instead of cross-attention). The paper lacks experimental validation to confirm if the proposed approach is applicable and effective for these transformer-based diffusion models.
- While the ablation studies are extensive, the results indicate significant sensitivity to hyperparameter tuning for the loss components. This raises concerns about the method's generalizability, as it may require extensive, model-specific, or test dataset-specific hyperparameter searching to achieve optimal performance, potentially undermining its plug-and-play nature.

**Questions:**

- The method introduces loss calculations at every sampling step, which presumably impacts computational overhead. Could the authors quantify the additional latency (e.g., percentage increase in sampling time) introduced by this process? Given that diffusion sampling is already computationally intensive, significant overhead could be a practical disadvantage.
- The introduction states that “state-of-the-art performances on spatial understanding are
around 20%, significantly lagging behind performance on other aspects such as attribute binding
(around 50%).” Could the authors please clarify which specific SOTA models and benchmarks this figure refers to? As far as I am aware, recent models (e.g., SD3.5, QwenImage, Infinity) appear to demonstrate significantly stronger text-image alignment.

---

### Official Review · Reviewer_8KSx · 2025-11-03

**Soundness:** 3
**Presentation:** 3
**Contribution:** 2
**Rating:** 4
**Confidence:** 3

**Summary:**

This paper proposes InfSplign, a training-free, inference-time method that improves the spatial alignment of objects in text-to-image generation by adjusting the noise at each denoising step via a compound loss. The method leverages hierarchical cross-attention maps from the U-Net decoder to enforce accurate object placement and balanced presence. Evaluations on spatial benchmarks (VISOR and T2I-CompBench) show that InfSplign outperforms existing inference-time methods and even some fine-tuning-based approaches.

**Strengths:**

+ The method is training-free and requires no extra inputs, making it easy to deploy and compatible with various diffusion backbones.

+ It achieves competitive results on standard benchmarks, significantly outperforming other inference-time methods.

+ The hierarchical use of attention maps and the introduction of variance as a measure of uncertainty are thoughtful design choices.

**Weaknesses:**

- The method introduces several hyperparameters (e.g., α, m, λs, λp, λb, η). While a grid search was conducted, their generalizability and robustness across different datasets or models are not fully verified, potentially affecting usability.

- The approach primarily focuses on binary spatial relationships between two objects. Its applicability to more complex spatial layouts or scenes with multiple objects is not deeply explored, limiting its scope.

- The experiments are conducted exclusively on U-Net-based architectures (Stable Diffusion v1.4/v2.1). There is no validation on newer, transformer-based diffusion architectures like SD3 or Flux, leaving the method's effectiveness and generalizability across different model architectures unproven.

**Questions:**

- How would the InfSplign loss function be extended for text prompts containing more than two objects requiring precise spatial layout? Is the current triplet formulation 〈A,R,B〉 sufficient for handling more complex spatial descriptions?

- The paper mentions performance limitations with uncommon object combinations. Beyond object omission, does InfSplign potentially sacrifice image plausibility or aesthetic quality to satisfy spatial constraints in such cases? Are there any related observations or metrics?

- The guidance weight η is set to a large, fixed value (1000). What was the rationale for this specific choice? Is the performance sensitive to this parameter across different prompts or random seeds? Is there evidence of robustness?

---

### Note · Authors · 2025-11-14

**Comment:**

We would like to thank the reviewers for their insightful and constructive feedback. We do appreciate the time and effort invested in evaluating our submission.

We continue to believe that our proposed approach ($\texttt{InfSplign}$) and the accompanying loss formulation are novel and represent a meaningful step forward in advancing the current state of the art in spatial cognizance of diffusion models. Our proposed method offers significant performance gains on the most widely adopted spatial cognizance benchmarks, **VISOR** and **T2I CombBench**. We would like to highlight that the focus on the four primary spatial relationships ($\textit{on top of}$, $\textit{at the bottom of}$, $\textit{to the left of}$, and $\textit{to the right of}$) stems from the fact that nearly all current state-of-the-art baselines employ these same relationships, and these two prominent benchmarks exclusively evaluate them.

While we have already prepared additional results for more recent diffusion models, such as SDXL, for the rebuttal, we believe that extending our work to Transformer-based architectures or to a broader set of spatial relationships requires further development that cannot be meaningfully addressed within the rebuttal period. In light of this, we have decided to withdraw the paper at this time.
We once again thank the reviewers for their valuable comments, which will undoubtedly help us strengthen the work for future submission.

**Withdrawal Confirmation:**

I have read and agree with the venue's withdrawal policy on behalf of myself and my co-authors.